



**Characterization of the MISG soot generator with an atmospheric simulation chamber**
*Virginia Vernocchi[1,2], Marco Brunoldi[1,2], Silvia G. Danelli[1,2], Franco Parodi[2], Paolo Prati[1,2], Dario*
*Massabò[1,2,\*]*
[1]*Dipartimento di Fisica - Università di Genova, via Dodecaneso 33, 16146, Genova (Italy)*
[2]*INFN – Sezione di Genova, via Dodecaneso 33, 16146, Genova (Italy)*
*\*Correspondence to: D. Massabò (massabo@ge.infn.it)*
**ABSTRACT**
The performance of a Mini-Inverted Soot Generator (MISG) has been investigated at ChAMBRe (Chamber
for Aerosol Modelling and Bio-aerosol Research) by studying the properties of soot particles generated by
ethylene and propane combustion.
Starting from an extensive classification of combustion conditions and resulting flame shapes, the MISG
exhaust was characterized in terms of concentration of emitted particles and gases, particle size distribution
and optical properties. Soot particles were also collected on quartz fibre filters and then analysed by optical
and thermal-optical techniques, to measure the spectral dependence of the absorption coefficient b_abs, and
their composition in terms of Elemental and Organic Carbon (EC and OC). Significant differences could be
observed when the MISG is fuelled with ethylene and propane both in terms of particle size and optical
behaviour (i.e., absorption coefficient). Values of the Mass Absorption Coefficient (MAC) and of the
Angstrom Absorption Exponent (AAE) turned out to be compatible with the literature, even if with some
specific difference.
The comprehensive characterization of the MISG soot particles is an important piece of information to
design and perform experiments in atmospheric simulation chambers.
**1.  Introduction**
"Soot" refers to combustion-generated carbonaceous particles that are a by-product of incomplete
combustion of fossil fuels and/or biomass burning (Nordmann et al., 2013; Moore et al., 2014). When
investigated by optical techniques, soot particles are generally referred as Black Carbon, BC (Petzold et al.
2013) while the result of thermal - optical characterizations is referred as Elemental Carbon, EC, (Bond and
Bergstrom, 2006). However, both BC and EC are defined in operative terms that do not identify the same
compounds (Massabò and Prati, 2021) and often produce non-negligible differences in concentration values.
Soot particles constitute an important fraction of anthropogenic particulate matter (PM) especially in urban
environments (Weijer et al. 2011), and are emitted by traffic, domestic stoves, industrial chimneys and by any
incomplete combustion process.
Several works state adverse effects of soot both on climate (Ackerman et al., 2000; Menon et al., 2002;
Quinn et al., 2008; Ramanathan and Carmichael, 2008; Bond et al., 2013) and health (Pope et al., 2002;
Anenberg et al., 2010; Gan et al., 2011; Cassee et al., 2013; Lelieveld et al., 2015). From the climatic point of
view, soot particles absorb the solar radiation, causing a positive radiative forcing. Effects on health include
cardiopulmonary morbidity and mortality (Janssen et al., 2012). The understanding of properties and behaviour
of soot particles when they are suspended in the atmosphere is thus necessary to fully assess their adverse
effects and the use of proxies with controlled and known properties can be useful.
So far, soot generators have been employed for studies on optical properties (Zhang et al. 2008; Cross et
al. 2010; Mamakos et al. 2013; Utry et al. 2014 b; Bescond et al. 2016), instruments calibration (Onasch et al.
2012; Durdina et al. 2016) and several other purposes (Pagels et al. 2009; Henning et al. 2012; Ghazi et al.
2013; Ghazi and Olfert 2013). The Inverted-Flame Burner (Stipe et al. 2005) is often considered as an ideal
soot source (Moallemi et al., 2019 and references therein), due to its capacity to generate almost pure-EC
particles and for the stability of the flame and of its exhaust (Stipe et al. 2005). To such category belongs the





Mini-Inverted Soot Generator, MISG (Argonaut Scientific Corp., Edmonton, AB, Canada, Model MISG–2),
used in this work.
The MISG can be operated with different fuels: ethylene (Kazemimanesh et al., 2019), propane (Moallemi
et al., 2019), and theoretically also with ethane or fuel blends with methane and nitrogen, even if, to our
knowledge, no literature is available on such configurations. The air to fuel flow ratio can be adjusted to control
concentration and size of the generated particles. The maximum reachable concentration is about $10^7$ particles
$cm^{-3}$ (https://www.argonautscientific.com/), while particle size ranges from few tens to few hundreds of nm.
The behaviour of soot particles can be efficiently studied in/by atmospheric simulation chambers (ASCs):
these are exploratory platforms which allow to study atmospheric processes under controlled conditions, that
can be maintained for periods long enough to reproduce realistic environments and to study interactions among
their constituents (Finlayson - Pitts and Pitts, 2000; Becker, 2006). Recent examples concern the investigation
of the optical properties of mineral dust (Caponi et al., 2017) and wood-burning exhausts (Kumar et al., 2018).
Coupling the MISG to an ASC makes possible systematic experiments on the properties of soot particles
exposed and maintained in different conditions. In this work we mainly investigated the differences between
MISG exhausts produced by ethylene and propane burning.

**2.   Materials and methods**
**2.1   Mini-Inverted Soot Generator**

The MISG, introduced by Kazemimanesh (2019), is a combustion-based soot generator working as an
inverted-flame burner (Stipe et al., 2005) where air and fuel flow in an opposite way to the buoyancy force of
the hot exhaust gases. This results in a co-flow diffusion flame and leads to a better flame stability by reducing
flame tip flickering (Kirchstetter & Novakov, 2007; Stipe et al., 2005) and consequently to a more stable soot
particle generation.
The MISG is fed with air and fuel supplied by specific cylinders: we used both ethylene and propane, two
fuels with a well-known capability of producing soot. Air and fuel flow rates are controlled by two mass flow
controllers (MFCs, Bronkhorst High-Tech B.V., Ruurlo, Netherlands, Models F-201CV-10K-MGD-22-V and
FG-201CV-MGD-22-V-AA-000, respectively) operated via a home-made National Instruments Labview
code. The air and fuel flows can be controlled in the range 0-12 lpm and 0-200 mlpm, respectively. Differently
from other commercial generators, the MISG does not require a third gas (i.e., $N_2$) used as a carrier and the air
flow is internally split between combustion and carriage operations. This implies that the ratio of comburent
and carrier gas is not controllable, and the user can only adjust the comburent to fuel ratio.
The efficiency of the combustion process can be given in terms of the global equivalence ratio, starting
from the air-to-fuel ratio (AFR):

$$AFR = \frac{m_A}{m_F} = \frac{n_A * M_A}{n_F * M_F}$$
where:
$m_A$: air mass;
$m_F$: fuel mass;
$n_A$: number of air moles;
$n_F$: number of fuel moles;
$M_A$: air molecular weight;
$M_F$: fuel molecular weight.





The stoichiometric AFR value is 15.64 $m^3$ $m^{-3}$ (inverse value = 0.064 $m^3$ $m^{-3}$) and 14.75 $m^3$ $m^{-3}$ (inverse value
= 0.068 $m^3$ $m^{-3}$), for propane and ethylene, respectively. Finally, the ratio between stoichiometric and actual
AFR corresponds to the global equivalence ratio:


$$\varphi = \frac{(m_F / m_A)}{(m_F / m_A)_{st}}$$


where:

$(m_F/m_A)$: inverse of actual AFR;

$(m_F/m_A)_{st}$: inverse of stoichiometric AFR.


The flame is classified as fuel-rich and fuel-lean when $\phi > 1$ and $\phi < 1$, respectively. It is yet demonstrated
(Moore et al., 2014) that fuel-lean flames produce soot particles with larger mode diameter (about 100-200
nm) while fuel-rich flames lead to an additional mode in the nucleation size range (i.e., 10-30 nm). Finally,
Mamakos (2013) reported that low fuel-to-air ratios (i.e., $\phi < 1$) generate particles with a large fraction of EC
while semi-volatile organics are generated by high fuel-to-air ratios (i.e., $\phi > 1$). In this work, fuel-lean
conditions were investigated only.

Since the combustion process can produce flame shapes having different characteristics, we first explored

the range of combustion flows from 2 to 10 lpm, in 0.5 lpm steps, and from 30 to 100 mlpm, in 5 mlpm steps,
respectively for air and fuel. Flame types can be distinguished (Kazemimanesh et al., 2019; Moallemi et al.,
2019) as:
- *Closed tip* flame (Fig. 1.a), which generates low concentrations of soot particles (i.e., around $10^3$ # $cm^{-3}$),
generally forming particle aggregates at the nozzle of the MISG.
- *Partially Open tip* flame (Fig. 1.b), the transition between *Open* and *Closed tip*.
- *Open tip* flame (Fig. 1.c), which generates high concentrations of soot particles (i.e., > $10^5$ # $cm^{-3}$).
- Asymmetric flame, which shows a large variability (very short, flickering, etc) and can form particle
aggregates at the MISG nozzle.
- *Curled Base* flame (Fig. 1.d), a particular shape of the asymmetric flames that can also form particles
aggregates at the MISG nozzle.

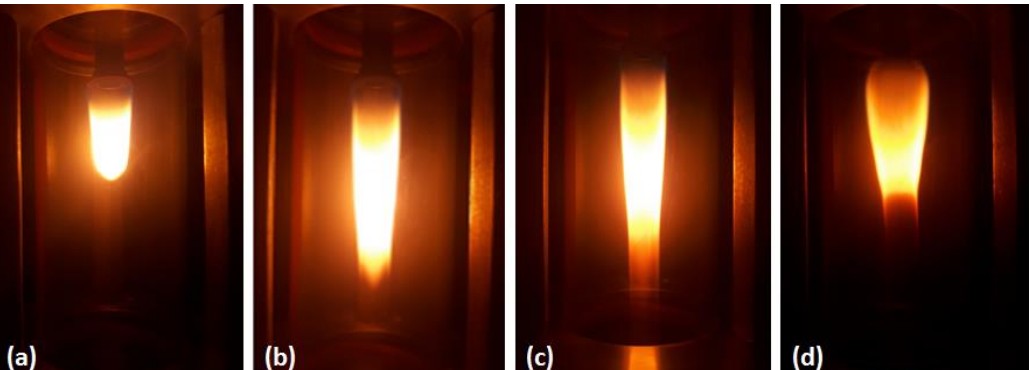

*Figure 1: Examples of different flame shapes: (a) Closed tip, (b) Open tip, (c) Partially Open tip, (d) Curled base flame.*

By the flames observation (Sect. 2.1.2), we selected the more interesting combustion conditions (i.e., *Open*

*tip* flames) to perform the characterization experiments. We focused on *Open tip* flames because it is the flame
that generates higher concentrations of soot particles. Operative conditions selected for propane and ethylene
combustion are reported in Tables 1 and 2: we maintained the same air flow and global equivalence ratio with
both the fuels.
*Table 1: Combustion parameters and flame shapes selected for propane.*

| PROPANE | | | |
|---|---|---|---|
| AIR flow [lpm] | FUEL flow [mlpm] | Global Equivalence Ratio | Flame shape |
| 7 | 70 | 0.244 | Partially Open Tip |
| 7 | 75 | 0.261 | Open Tip |
| 7 | 80 | 0.278 | Open Tip |
| 7 | 85 | 0.296 | Open Tip |
| 8 | 70 | 0.213 | Partially Open Tip |
| 8 | 75 | 0.228 | Open Tip |
| 8 | 80 | 0.244 | Open Tip |
| 8 | 85 | 0.259 | Open Tip |

*Table 2: Combustion parameters and flame shapes selected for ethylene.*

| ETHYLENE | | | |
|---|---|---|---|
| AIR flow [lpm] | FUEL flow [mlpm] | Global Equivalence Ratio | Flame shape |
| 7 | 118 | 0.244 | Partially Open Tip |
| 7 | 127 | 0.261 | Open Tip |
| 7 | 135 | 0.278 | Open Tip |
| 7 | 144 | 0.296 | Open Tip |
| 8 | 118 | 0.213 | Partially Open Tip |
| 8 | 127 | 0.228 | Open Tip |
| 8 | 135 | 0.244 | Open Tip |
| 8 | 144 | 0.259 | Open Tip |



**2.2 Chamber setup**
Experiments took place at the ChAMBRe (Chamber for Aerosol Modelling and Bio-aerosol Research)
facility (Massabò et al., 2018; Danelli et al., 2021) located at the Physics Department of the University of
Genoa.





ChAMBRe is a stainless-steel chamber, with a volume of about 2.2 m³. Inside the chamber, relative
humidity, temperature, and pressure are continuously monitored by a HMT334 Vaisala® Humicap®
transmitter and a MKS Instruments 910 DualTrans™ transducer, respectively. Two gas analyzers from
Environnement SA, continuously monitored the concentration of $NO/NO_2$ (model: AC32e), and $CO/CO_2$
(model: CO12e) inside the chamber or, alternatively, in the laboratory. The mixing of gas and aerosol species
is favoured by a fan installed in the bottom of the chamber: mixing time for gaseous species is of about 180 s
with a fan rotating speed of 1.6 revolutions per second. A composite pumping system (rotary pump TRIVAC®
D65B, Leybold Vacuum, root pump RUVAC WAU 251, Leybold Vacuum and Leybold Turbovac 1000)
allows to evacuate the internal volume down to $10^{-5}$ mbar; in this way ChAMBRe is cleaned before each
experiment. Before and during the experiments, ambient air enters the chamber throughout a 5-stage
filtering/purifying inlet (including a HEPA filter, model: PFIHE842, NW25/40 Inlet/Outlet – 25/55 SCFM,
99.97 % efficient at 0.3 μm). The whole set-up is managed by a custom NI Labview SCADA (Supervisory
Control And Data Acquisition).
The layout of the experimental configuration adopted for the MISG characterization is shown in Fig. 2.
The MISG was warmed for about 45 minutes before injecting soot particles inside the chamber. Injection
of soot particles inside ChAMBRe lasted 2 or 3 minutes, depending on the soot concentration required for each
experiment. We performed some fluid dynamic evaluations with the Particle Loss Calculator software tool
(PLC; von der Weiden et al., 2009). The geometry of our experimental setup, combined with particle size and
used flow rates, resulted in particle losses lower than 0.1 % in the dimensional range of 80-2000 nm. All the
experiments were performed at atmospheric pressure, 19° < T< 21 °C and R.H. < 50 %.

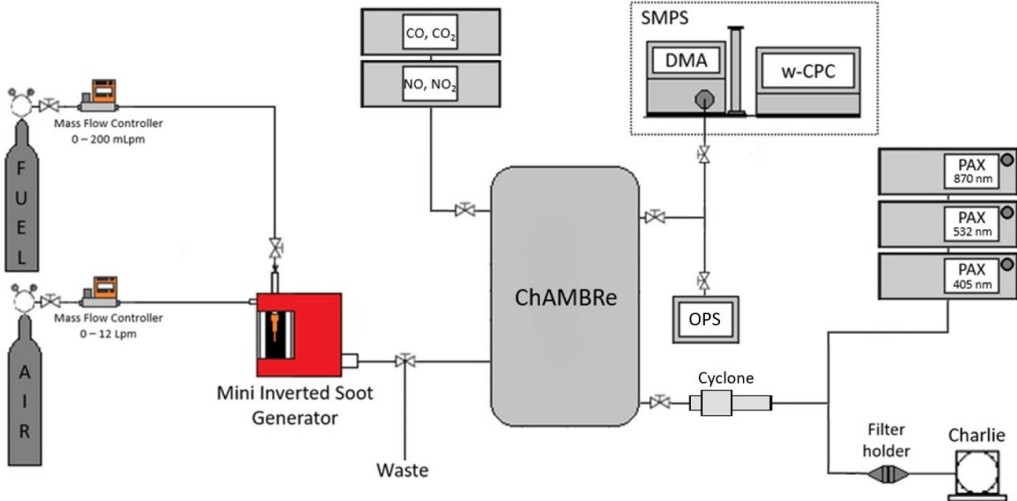

*Figure 2: Layout of the MISG set-up at ChAMBRe.*

**2.3 Size distributions**
Particle concentration and size distribution inside the chamber were measured by a scanning mobility
particle sizer (SMPS, TSI Inc., Shoreview, MN, USA, Model 3938), composed by a differential mobility
analyzer (DMA, TSI Inc., Shoreview, MN, USA, Model 3081A) and a water condensation particle counter
(w-CPC, TSI Inc., Shoreview, MN, USA, Model 3789). The water-CPC is filled using technical demineralized
water (Conductivity (20°C), max. 1.5 μS/cm; VWR Chemicals INTERNATIONAL S.R.L.). The SMPS was





set to measure particles with mobility diameter from 34 nm to 649 nm; aerosol sample and sheath airflow rates
were fixed at 0.17 lpm and 1.60 lpm, respectively, while the scanning period for each cycle was 70 s. The
DMA unit integrates an impactor with an orifice of 0.0508 cm, resulting in cut-off capability at 50 % of 940
nm, useful to exclude all the particles larger than this size to enter in the column. Frequent cleaning of this part
was necessary to ensure proper operation and avoid clogging; at the end of each experiment, the whole
impactor system was cleaned using compressed air and isopropyl alcohol.
We corrected diffusion losses in the instrument by using the option included in the instrument software;
size distributions were as well corrected for multiple charges effects through the TSI proprietary software
(Aerosol Instrument Manager, Version 11-0-1).
Among the other chamber instruments, an Optical Particle Sizer (OPS, TSI Inc., Shoreview, MN, USA,
Model 3330) was used for short times to spot the particle size distribution in the range 0.3-10 µm.

**2.4 Online optical measurements**
Three photoacoustic extinction-meters (PAXs, Droplet Measurement Technologies, Boulder, CO, USA)
were deployed, providing the online determination of the soot particles absorption coefficients at $\lambda = 870$, 532
and 405 nm. PAXs are constituted by a measurement cell where aerosol optical properties are measured by
two different mechanisms (https://www.dropletmeasurement.com/ PAX Operator Manual). The sample flow
rate (1 lpm) is split in two different sectors of the cell, both crossed at the same time by the light of a modulated
laser diode. In the absorption sector, soot particles absorb light and release acoustic waves which are then
detected by an ultra-sensitive microphone. The intensity of the acoustic signal is interpreted to infer the particle
absorption coefficient. In the other sector, a wide angle reciprocal nephelometer measures the scattering
coefficient instead. It is noteworthy that no correction for the truncation angle is applied by the manufacturer:
this can lead to substantial underestimation of the scattering coefficient, which generally grows as the particle
size increases and the single scattering albedo (SSA) approaches unity. However, since particles produced by
soot generators have dimensions much lower than 1 µm and SSA values lower than 0.3 (Moallemi et al., 2019),
we disregarded this issue. The PAXs had been calibrated by the manufacturer.
In some experiments, soot concentration inside the chamber was too high to be measured directly by PAXs;
and a diluter (eDiluter Pro, Dekati Ltd., Kangasala, Finland) was deployed. Dry air from a cylinder was merged
prior to the PAXs inlet with dilution factor 1:100. Tests performed with and without the diluter demonstrated
a substantial reproducibility of the optical properties measured by the PAXs when the proper dilution factor is
considered.

**2.5 Offline analysis**
Soot particles were also collected on pre-fired 47 mm diameter quartz fibre filters (Pallflex Tissuquartz
2500 QAO-UP) held in a stainless-steel filter holder to allow additional offline analysis. The sampling started
when stable gas and particle concentration values were reached inside the chamber (i.e., about 3 minutes -
corresponding to the chamber mixing time - after the MISG switching off): for each working condition three
filters with different loadings were obtained by a low-volume sampler (TECORA – Charlie HV) working at a
fixed sampling flow (i.e., 10 lpm during experiments without cyclone and 13.67 lpm during experiments with
cyclone).
For each sample, the EC and OC mass concentration was determined by thermal-optical transmittance
analysis (TOT) using a Sunlab Sunset EC/OC analyzer and the NIOSH5040 protocol (NIOSH, 1999),
corrected for temperature offsets.
Prior to EC/OC determination, particle-loaded filters were analyzed by the Multi-Wavelength Absorbance
Analyzer (MWAA, Massabò et al., 2013 and 2015), a laboratory instrument for the offline direct quantification
of the aerosol absorption coefficients at five different wavelengths ($\lambda = 850$, 635, 532, 405 and 375 nm). Such
peculiar feature had been previously exploited in the frame of several field campaigns in urban and rural sites





(Scerri et al., 2018; Massabo et al, 2019; Massabo et al, 2020; Moschos et al., 2021), as well as in peculiar and
remote sites (Massabo et al., 2016; Saturno et al., 2017; Baccolo et al., 2021).
**2.6 Cyclone experiments**
Soot aggregates are also generated by the MISG. Kazemimanesh (2019) retrieved super-aggregates larger
than 2 μm for ethylene combustion while Moallemi (2019) showed aggregate structures larger than 1 μm with
propane. On this basis, confirmed by some short checks by the OPS, we replicated each experiment (see Sect.
2.1) both without and with a cyclone (PM1 Sharp Cut Cyclone - SCC 2.229, MesaLabs, Lakewood, CO, USA)
inserted upstream the PAXs and filters sampler (Fig. 2). The cyclone has a cut-off of 1 μm at a nominal flow
of 16.66 lpm.
**3. Results and Discussion**
**3.1 Characterization tests**
The categories of flame shape observed in the range of air and fuel flows discussed in sect. 2.1.2 are
summarized in Tables 3 and 4, for propane and ethylene respectively. The MISG characterization with propane
has been previously published (Moallemi et al., 2019) and we used it as reference. We got some differences
especially in the range of transition from *Closed tip* to *Open tip*, probably due to the different setups. Fuel
flows higher than 85 mlpm were not investigated due to instrumental limitation. To our knowledge, no
literature information is available for the ethylene in the flow range of Table 4. It is noteworthy that no
correlation could be found between the global equivalence ratio ($\phi$) and the shape of the corresponding flame.
This means that the fundamental parameter of the combustion process can not be used to predict the flame
shape.
The reproducibility and stability of the MISG emissions were investigated, in terms of number
concentration and size distribution of the generated soot particles. Different combustion conditions were
selected, and four experiments were performed for each combination of air and fuel flows. We chose to keep
fixed the air flow to observe the differences produced by different fuel flows that correspond to different flame
shapes (i.e., *Partially Open tip* or *Open tip*). In each test, we recorded the values of total particle number
concentration, peak concentration, and mode diameter. The reproducibility was calculated as the percentage
ratio between standard deviation and mean value of each series of repeated experiments. With propane, mode
reproducibility turned out to be 6 %, while total concentration and peak concentration showed a 16 %
reproducibility. With ethylene, the reproducibility was 4 % and 10 %, respectively for mode and total/peak
concentration. In addition, we monitored the combustion gases: $CO_2$ and $NO$ concentration varied by about 2
% and 3 %, respectively with propane and ethylene.

*Table 3: Flame shapes observed for different combustion conditions of propane. Flames are identified as A - asymmetric,*
*CB - Curled Base, CT - Closed tip, POT - Partially Open tip and OT - Open tip; FL indicates if flickering. The dash*
*indicates that the flame does not ignite.*





| FUEL flow [mlpm] | | | | | | | | | | | | |
|---|---|---|---|---|---|---|---|---|---|---|---|---|
| | 30 | 35 | 40 | 45 | 50 | 55 | 60 | 65 | 70 | 75 | 80 | 85 |
| 2 | A | A/FL | A | A/FL | A | CB/FL | CB/FL | CB/FL | CB/FL | CB/FL | CB/FL | CB/FL |
| 2.5 | A/FL | A/FL | A | A/FL | A/FL | CB/FL | CB/FL | CB/FL | CB/FL | CB | CB | CB/FL |
| 3 | A/FL | A/FL | A | A/FL | A/FL | CB/FL | CB/FL | CB/FL | CB/FL | CB/FL | CB | CB/FL |
| 3.5 | A/FL | A | A | A | A | CB | CB | CB | CB | CB | CB | CB/FL |
| 4 | A | A | A | A | A | CB | CB | CB/FL | CB | CB | CB | OT |
| 4.5 | A | A | A | A | A | CB | CB | CB | CB | CB | CB | OT |
| 5 | A | A | A | A | A | A/CB | CT | POT | OT | OT | OT | OT |
| 5.5 | A | A | A | A | A | CT | CT | POT | OT | OT | OT | OT |
| 6 | A | A | A | A | CT | CT | CT | CT | POT/OT | OT | OT | OT |
| 6.5 | A | A | A | A | CT | CT | CT | CT | POT | OT | OT | OT |
| 7 | A | A | A | A | A | CT | CT | CT | POT | POT/OT | OT | OT |
| 7.5 | A | A | A | A | A | CT | CT | CT | POT | POT/OT | OT | OT |
| 8 | - | - | A | A | A | CT | CT | CT | POT | POT/OT | OT | OT |
| 8.5 | - | - | A | A | A | CT | CT | CT | POT/OT | POT/OT | OT | OT |
| 9 | - | - | A | A | A | CT | CT | CT | CT | POT | OT | OT |
| 9.5 | - | - | - | A | A | CT | CT | CT | CT | POT | OT | OT |
| 10 | - | - | - | A | A | CT | CT | CT | CT | POT | OT | OT |

AIR flow [lpm]

Table 4: Flame shapes observed for different combustion conditions of ethylene. Flames are identified as A - asymmetric, CB - Curled Base, CT - Closed tip, POT - Partially Open tip and OT - Open tip; FL indicates if flickering.

| FUEL flow [mlpm] | | | | | | | | | | | | | | | |
|---|---|---|---|---|---|---|---|---|---|---|---|---|---|---|---|
| | 30 | 35 | 40 | 45 | 50 | 55 | 60 | 65 | 70 | 75 | 80 | 85 | 90 | 95 | 100 |
| 2 | A | A | A | A | A/FL | A | A | A | CB | CB | CB | CB | CB | CB | CB |
| 2.5 | A | A | A | A/FL | A/FL | A/FL | A | A | CB | CB | CB | CB | CB | CB | CB |
| 3 | A | A | A | A | A/FL | A/FL | A | A | A/CB | CB | CB | CB | CB | CB | CB |
| 3.5 | A | A | A | A | A/FL | A/FL | A | A | A/CB | CB | CB | CB | CB | CB | CB |
| 4 | A | A | A | A | A | A | A | A | A | A/CB | CB | CB | CB | CB | CB/OT |
| 4.5 | A | A | A | A | A | A | A | A | CB | CB | CB | CB | CB/OT | CB/OT | CB/OT |
| 5 | A | A | A | A | A | A | A | A | CB | CB | CB | CB | CB/OT | CB/OT | CB/OT |
| 5.5 | A | A | A | A | A | A | A | A | CB | CB/OT | CB/OT | CB/OT | CB/OT | CB/OT | CB/OT |
| 6 | A | A | A | A | A | A | CT | CT | CT | CT/POT | CT/POT | POT | POT | OT | OT |
| 6.5 | A | A | A | CT | CT | CT | CT | CT | CT/POT | POT | POT/OT | POT/OT | OT | OT | OT |
| 7 | A | A | A | CT | CT | CT | CT | CT/POT | POT | POT/OT | OT | OT | OT | OT | OT |
| 7.5 | A | A | A | A | CT | CT | CT | CT | POT | POT/OT | OT | OT | OT | OT | OT |
| 8 | A | A | A | CT | CT | CT | CT | CT/POT | POT | POT/OT | OT | OT | OT | OT | OT |
| 8.5 | A | A | A | CT | CT | CT | CT | CT | CT/POT | POT | OT | OT | OT | OT | OT |
| 9 | A | A | CT | CT | CT | CT | CT | CT | POT | OT | OT | OT | OT | OT | OT |
| 9.5 | A | A | CT | CT | CT | CT | CT | CT | CT | POT | OT | OT | OT | OT | OT |
| 10 | A | CT | CT | CT | CT | CT | CT | CT/POT | POT | POT/OT | OT | OT | OT | OT | OT |

AIR flow [lpm]








### 3.2 Comparison between propane and ethylene exhausts

Previous works investigated concentration and mode of ethylene (Kazemimanesh et al., 2019) propane (Moallemi et al., 2019) fuelled MISG exhausts. We expand here to a detailed comparison between the two fuels, focusing on ASC experiments.

### 3.2.1 Size distribution

To compare different experiments, particle concentration values were normalized to the maximum recorded in the whole set of tests and therefore varied in the 0-1 range. Fig. 3 shows the result for the total particle number concentration, we can notice that:
- At fixed air flow, the particle number concentration increases with the fuel flow (i.e., with the global equivalence ratio).
- In the same combustion conditions (i.e., same air flow and same global equivalence ratio), ethylene generates more particles than propane.
- With ethylene and at fixed fuel flow, the particle number concentration increases with the air flow. The same holds in some cases with propane but with much smaller variations.

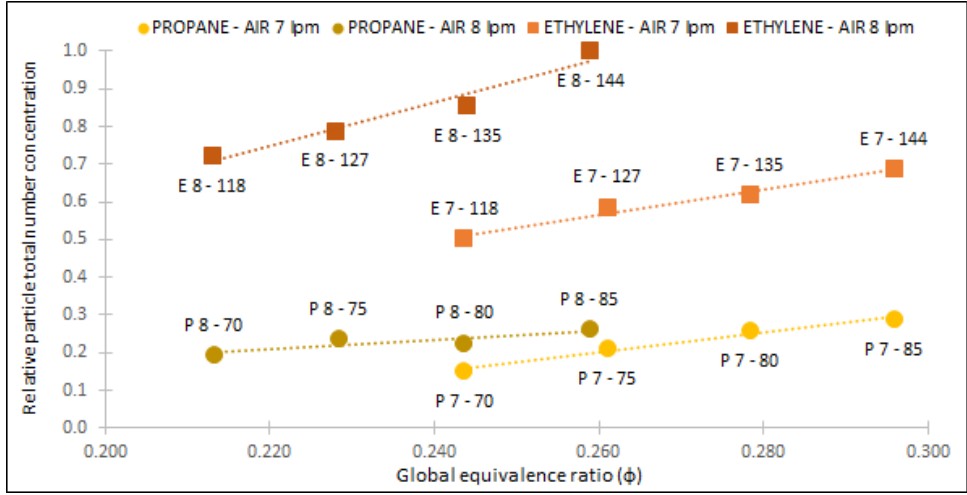

*Figure 3: Particle number concentration vs the global equivalence ratio. Values are normalized to the highest of the whole set. Each point is labelled by E or P (ethylene or propane) and a pair of numbers indicating air and fuel flow rate, respectively in lpm and mlpm.*

A similar comparison is shown in Fig. 4 for the particle mode diameter: while the values are basically constant for ethylene, the mode diameter with propane slightly increases with air flow (at fixed fuel flow). Furthermore, at each φ value, propane generated particles bigger than ethylene.





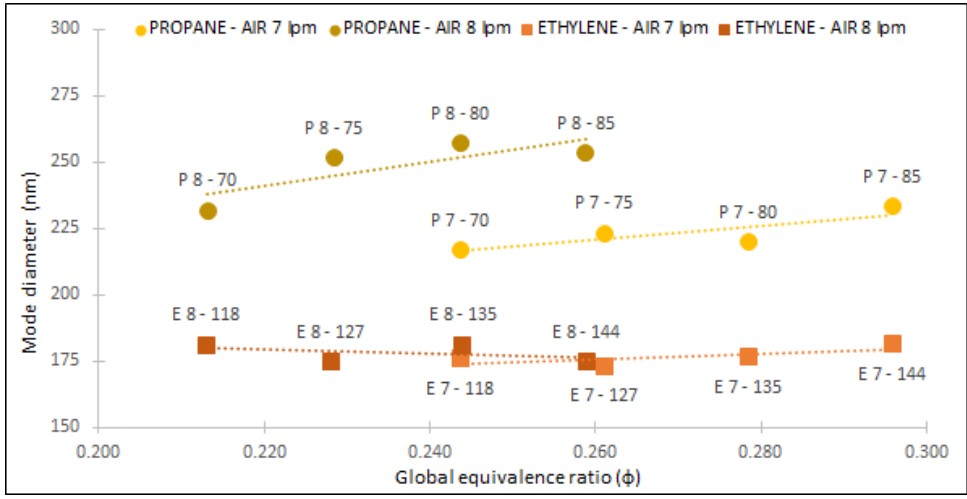

*Figure 4: Mode diameter versus the global equivalence ratio. Each point is indicated by E or P (ethylene or propane)*
*and a pair of numbers indicating air and fuel flow rate, respectively in lpm and mlpm.*

Both in (Kazemimanesh et al., 2019) with ethylene and (Moallemi et al., 2019) and (Bischof et al., 2019)
with propane, when the fuel flow increased, at a certain air flow, the particle number concentration increased
too; even if a lower range of global equivalence ratio were considered. In addition, Kazemimanesh (2019) and
Bischof (2019) also reported that the particle mode diameter, both with ethylene and propane, did not depend
on the global equivalence ratio, as we also observed. In (Moallemi et al., 2019), instead, they observed an
opposite behaviour for mode diameters: they retrieved that at fixed fuel flow, a higher air flow produced a
slight decrease of the mode diameter. Both (Moallemi et al., 2019) and (Bischof et al., 2019) measured mode
diameters < 200 nm, but this can be due to the specific combustion conditions (i.e., lower global equivalence
ratios resulting from higher air flow or lower fuel flow).

The mean size distributions observed at ChAMBre are given in Fig. 5, for all the selected operative
conditions. All the curves are normalized to the same injection time (i.e., 3 min).





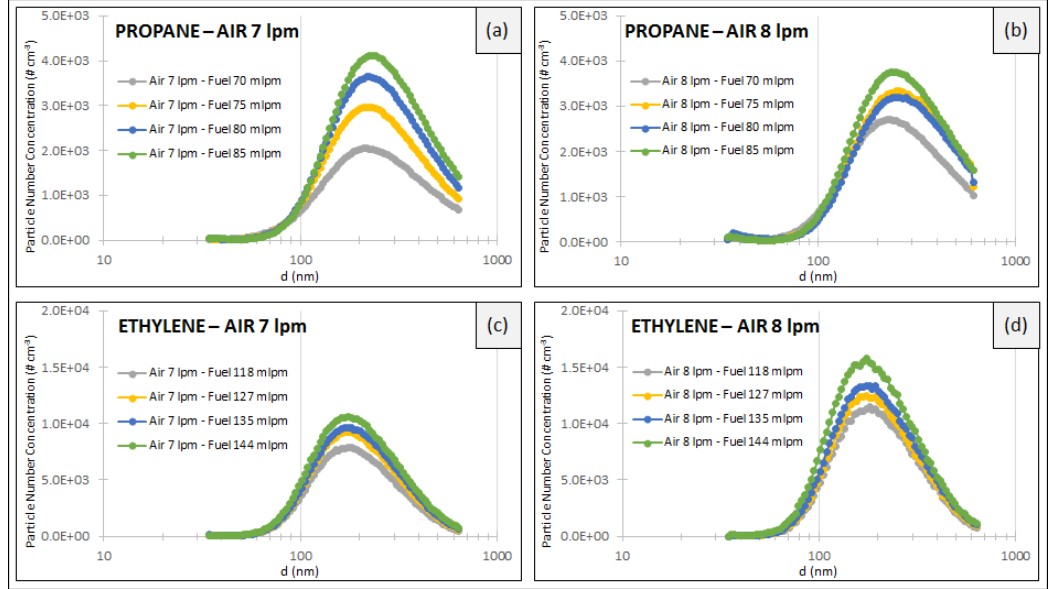

*Figure 5: Mean size distributions measured by SMPS. MISG was fuelled with propane (top panels) and ethylene (bottom panels) with the air and fuel flows indicated in the plots frame.*

Significant differences between the two fuels emerge when considering the particle mass concentration (extended to 10 μm, including the data collected by the OPS): ethylene combustion produced a limited number of big particles, likely super-aggregates, formed directly at the MISG exhaust. Kazemimanesh (2019) also observed the formation of aggregates, even with smaller dimensions (i.e., about 2 μm). We calculated the super-micrometric fraction of the total measured by the OPS with both the fuel (Fig. 6): this resulted to be about 3% with ethylene and 0.2% with propane. Particles larger than 4 μm were about 2% with ethylene, with a peak at 8 μm, and totally negligible with propane. Considering the particle volume distribution, the latter difference is obviously enhanced: the super-micrometric fraction is about 99% of the total concentration with ethylene and 9% only with propane. Particles larger than 4 μm contribute to the total volume (and hence to the soot concentration) for about 98% and 1%, respectively with ethylene and propane.



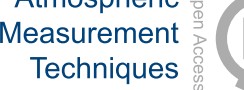

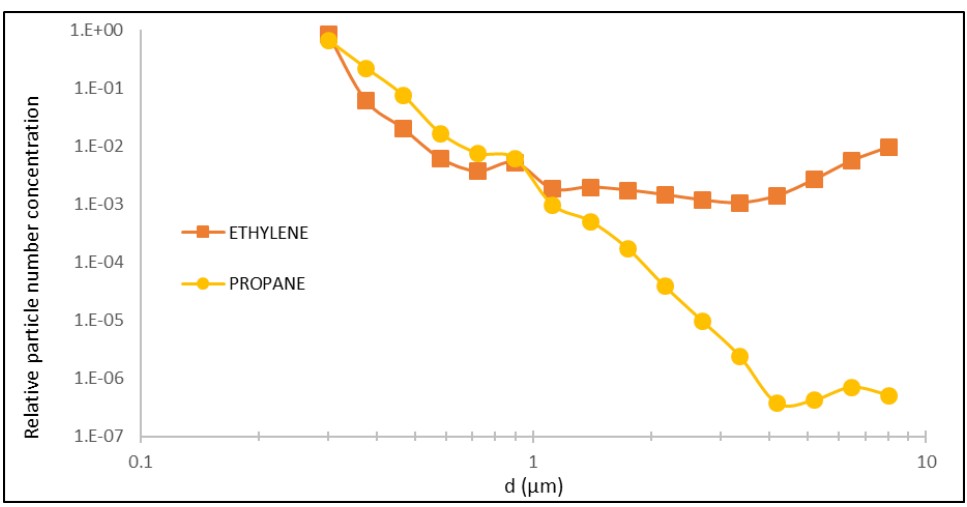

Figure 6: Relative particle number concentration normalized to the total number concentration of particles, in function
of particle size, measured by OPS.

### 3.2.2 Gaseous exhaust

Gaseous emissions were characterized too, focusing on the most abundant gases i.e., $CO_2$ and NO. The pattern is similar for both the gases: at fixed air flow rate, gas concentration increased with the fuel flow while no significant differences emerged at fixed fuel flow rate and changing the air flow. At equal operative conditions (i.e., same combustion conditions, injection time and time from the injection), gaseous emissions were higher with ethylene than with propane. With the same normalization introduced in Fig. 3, the $CO_2$ and NO production are compared in Fig. 7 and 8 for each selected MISG configuration. Maximum values were 360 ppm and 980 ppb, respectively for $CO_2$ and NO, after 3 minutes of soot injection.

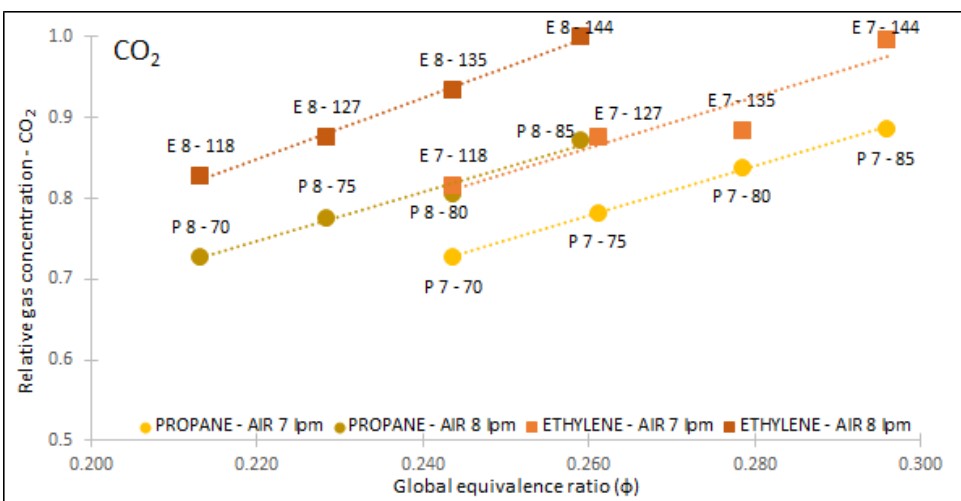


Figure 7: $CO_2$ concentration versus the global equivalence ratio. Each value was normalized to the highest of the whole set. Data points are labelled by E or P (ethylene or propane) and a pair of numbers indicating air and fuel flow, respectively in lpm and mlpm.





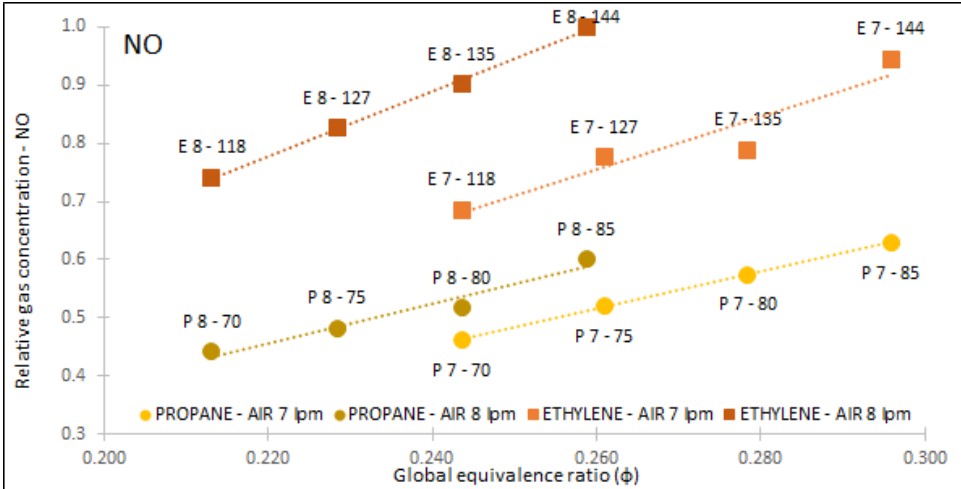


*Figure 8: NO concentration versus the global equivalence ratio. Each value was normalized to the highest of the whole*
*set. Data points are labelled by E or P (ethylene or propane) and a pair of numbers indicating air and fuel flow,*
*respectively in lpm and mlpm.*

### 3.2.3 EC/OC quantification


The OC/EC composition was quantified by thermal-optical analysis of samples collected on quartz fibre
filters during each experiment. EC:TC concentration ratios resulted to be around 0.7 and 0.9 with propane and
ethylene, respectively. In addition, the EC:TC concentration ratios increased with the global equivalence ratio.
All the results are given in Fig. 9 and 10, for experiments without and with cyclone respectively, adopting the
same normalization already introduced in Fig. 3. When removing large particles (see Sect 3.2.1), the EC:TC
concentration ratio resulted higher with propane (0.83 against 0.79 measured with ethylene). Actually, with
ethylene about 40 % of the EC concentration was associated with particles larger than 1 μm. With both fuels,
EC:OC ratios increase with the global equivalence ratios whether the cyclone is present or not, in agreement
with (Kazemimanesh et al., 2019) and (Moallemi et al., 2019).


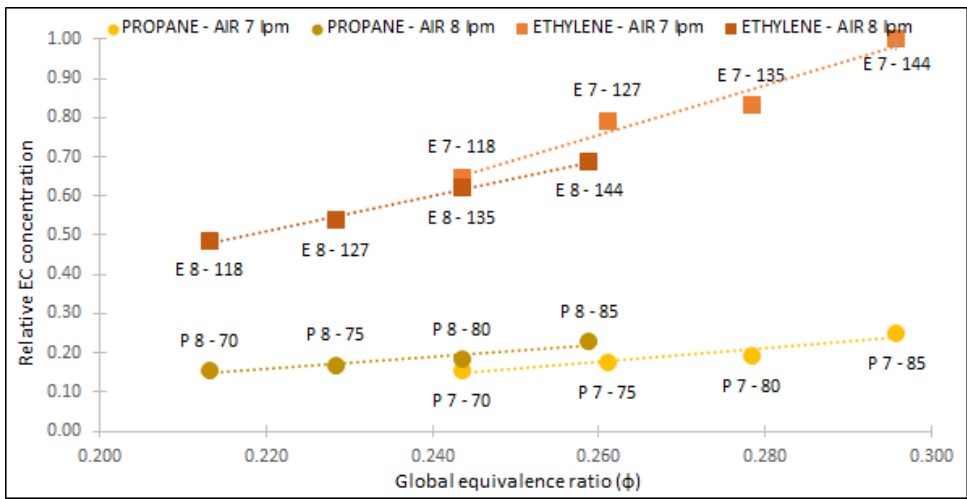


*Figure 9: EC mass concentration versus the global equivalence ratio, each value was normalized to the highest of the whole set. Each point is labelled by E or P (ethylene or propane) and a pair of numbers indicating air and fuel flow rate, respectively in lpm and mlpm. No cyclone.*


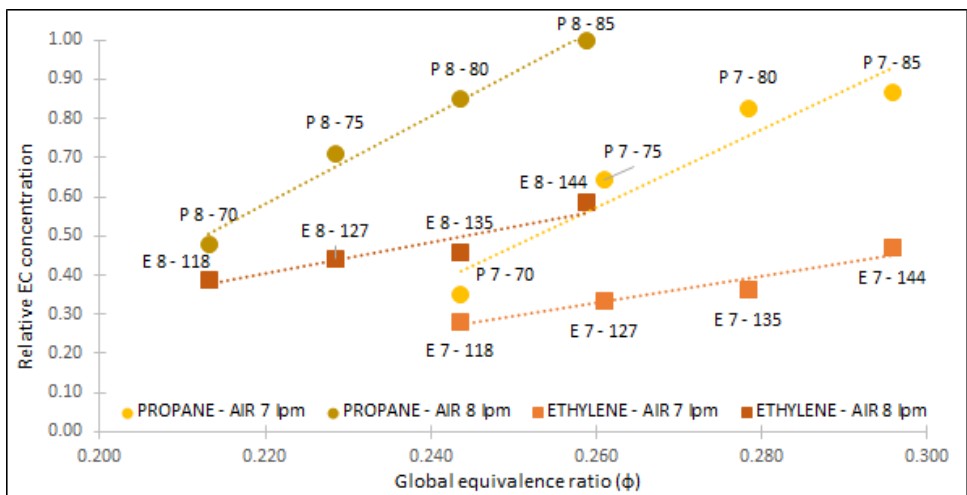


*Figure 10: EC mass concentration versus the global equivalence ratio, each value was normalized to the highest of the whole set. Each point is labelled by E or P (ethylene or propane) and a pair of numbers indicating air and fuel flow rate, respectively in lpm and mlpm. The cyclone impactor upstream the filter removed super-micrometric particles.*


The OC:EC ratio varies from 0.31 for propane to 0.19 and 0.10 for ethylene, with and without cyclone respectively. In each series of experiments (i.e., air flow rate 7 or 8 lpm, ethylene or propane) the OC fraction turned out to be inversely proportional to the fuel flow with a minimum at the lowest fuel flow (i.e., 70 lpm with propane and 118 lpm with ethylene). This is likely due to the shape of the flame: flames generated by the lowest fuel flow conditions are *Partially Open tip*, with less capability to generate soot particles and hence EC; so that the EC:OC ratio results lower.

We also performed some tests adding a backup filter during the sampling to catch the volatile fraction of OC. The OC concentration values measured on backup filters showed high variability, but they were compatible with those on not-sampled filters. We analysed 13 blank filters from different bunches and the





average concentration of OC resulted $<OC> = 0.5 \pm 0.2$ µg cm$^{-2}$ while OC concentration on backup filters was
$<OC_{BF}> = 0.6 \pm 0.2$ µg cm$^{-2}$ (average OC concentration on the corresponding main filters was $1.4 \pm 0.7$ µg
cm$^{-2}$). A relationship between OC concentration on the backup filter and the global equivalence ratio was
instead reported in (Kazemimanesh et al., 2019). Actually, in that study the range of investigated global
equivalence ratio values was $0.129 < \phi < 0.186$ to be compared $\phi > 0.210$ adopted in this work.

**3.2.4 Optical properties**
The optical properties of the MISG aerosol were determined in terms of the absorption coefficient (b_abs;
i.e. the absorbance per unit length) (Massabò and Prati, 2021). The b_abs definition applies both to
measurements directly performed on the aerosol dispersed in the atmosphere (by PAXs, in this work) and to
off-line analyses on aerosol sampled on filters (by MWAA, in this work), provided a proper data reduction is
adopted (Massabò and Prati, 2021; and references therein).
The measured b_abs values were normalized to the total particle concentration inside ChAMBRe reached
in each single experiment. Absorption coefficients measured at three wavelengths by the PAXs and with the
cyclone mounted upstream, are shown in Fig. 11. Similar results were obtained even for experiments without
cyclone and for the b_abs values measured by the MWAA. At each wavelength, the b_abs values did not show
any dependence on the global equivalence ratio, with the propane producing particles more absorbent than
ethylene. The comparison with previous literature (Moallemi et al., 2019) is hampered by methodological
differences (Moallemi and co-workers used the IR PAX only and reported the Single Scattering Albedo instead
of the absorption coefficient).






*Figure 11: Absorption coefficient @ λ = 870 (a), 532 (b) and 405 (c) nm, measured by PAXs, versus the global*
*equivalence ratio. b_abs values are normalized to the total particle number concentration measured by SMPS in the*
*corresponding experiments. Each point is labelled by E or P (ethylene or propane) and a pair of numbers indicating air*
*and fuel flow, respectively in lpm and mlpm.*

### 3.2.5 Mass Absorption Coefficient

The b_abs values, together with the EC concentration measured on the filter sampled during each single
experiment, can be used to retrieve the Mass Absorption Coefficient (MAC) of the produced aerosol, through
the relation:





$b\_abs(\lambda) = MAC * [EC]$

where:

$b\_abs$ [Mm$^{-1}$]: absorption coefficient

402          MAC [m$^2$ g$^{-1}$]: Mass Absorption Coefficient

403          EC [µg m$^{-3}$]: Elemental Carbon concentration

The $b\_abs$ values were calculated directly online by the PAXs and offline by the MWAA analysis,
performed at five wavelengths on the sampled filters (see Sect. 2.5). This gave the possibility to extend the
characterization of the MISG and to compare two optical analyses on the same carbonaceous aerosol. Since
experiments were repeated with two different setups (i.e., with and without the cyclone) and two different fuels
(propane and ethylene), four different particle populations can be compared. The comparison was carried out
at the three wavelengths (nearly) common to PAXs and MWAA (i.e., $\lambda$ = 870/850, 532 and 405 nm), as
reported in Fig. 11-13. We divided the results by fuel, air flow and with/without cyclone. Each point in the
plots sums-up the observations at different global equivalence ratio values.
The MWAA analysis at $\lambda$ = 870 nm (Fig. 12.a) returns compatible MAC values for both the propane series
(with/without cyclone) and the ethylene series with cyclone (average MAC = 5.25 ± 0.10 m$^2$ g$^{-1}$). At $\lambda$ = 532
and 405 nm (Fig. 13.a and Fig. 14.a), propane series are still in agreement while the ethylene series with
cyclone show a higher MAC value (MAC = 9.53 ± 0.08 m$^2$ g$^{-1}$ instead of MAC = 8.88 ± 0.13 m$^2$ g$^{-1}$ at $\lambda$ = 532
nm and MAC = 12.3 ± 0.1 m$^2$ g$^{-1}$ instead of MAC = 10.7 ± 0.2 m$^2$ g$^{-1}$ at $\lambda$ = 405 nm). The ethylene series
without cyclone show consistent variability at all the three wavelengths, with the lowest MAC values of the
whole data-set (MAC = 3.78 ± 0.08 m$^2$ g$^{-1}$; MAC = 5.9 ± 0.1 m$^2$ g$^{-1}$; MAC = 6.9 ± 0.1 m$^2$ g$^{-1}$ at $\lambda$ = 870, 532
and 405 nm respectively): the differences are probably due to the production of super-micrometric particles
(see Sect 3.2.1 and Fig. 6) when the cyclone is not used. With the PAXs analysis (Fig. 12.b, 13.b and 14.b),
MAC values turn out higher in the  series with cyclone (average values are  MAC = 5.8 ± 0.4 m$^2$ g$^{-1}$; 10.3 ±
0.1 m$^2$ g$^{-1}$; 14 ± 1 m$^2$ g$^{-1}$ at $\lambda$ = 870, 532 and 405 nm respectively, with cyclone, MAC = 4.3 ± 0.1 m$^2$ g$^{-1}$; 6.6
± 1.7 m$^2$ g$^{-1}$; 8 ± 2 m$^2$ g$^{-1}$ at $\lambda$ = 870, 532 and 405 nm respectively, without cyclone), this happens at all the
three wavelengths and for both fuels. If series with cyclone are only considered, MAC values do not show any
significant differences depending on the fuel. The ethylene series without cyclone shows the lowest MAC
values at each wavelength, as observed with the MWAA analysis. PAXs data show a higher variability in
MAC values, likely due to a higher sensitivity to particle size than filter based MWAA analysis.





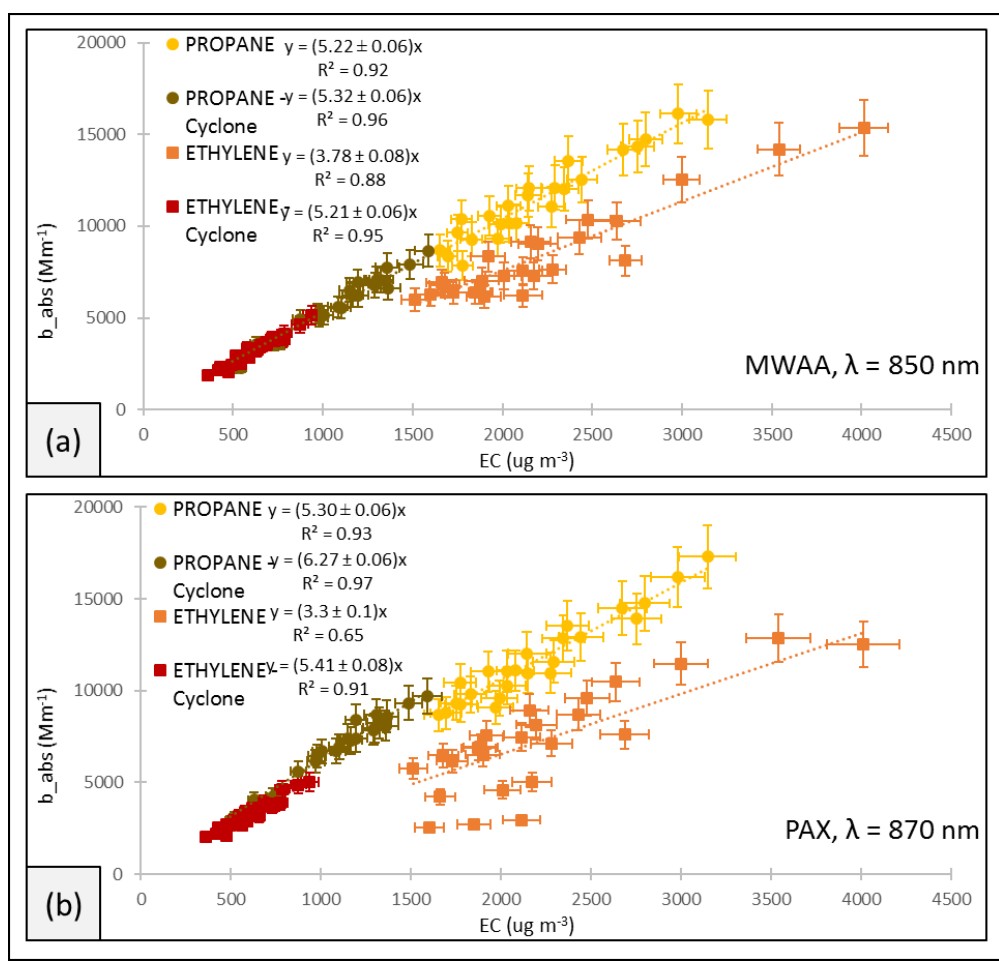


Figure 12: Absorption coefficient at 850 nm measured by MWAA (a) and @ 870 nm measured by PAX (b) versus EC
concentration. The slope of each fit corresponds to the Mass Absorption Coefficient.






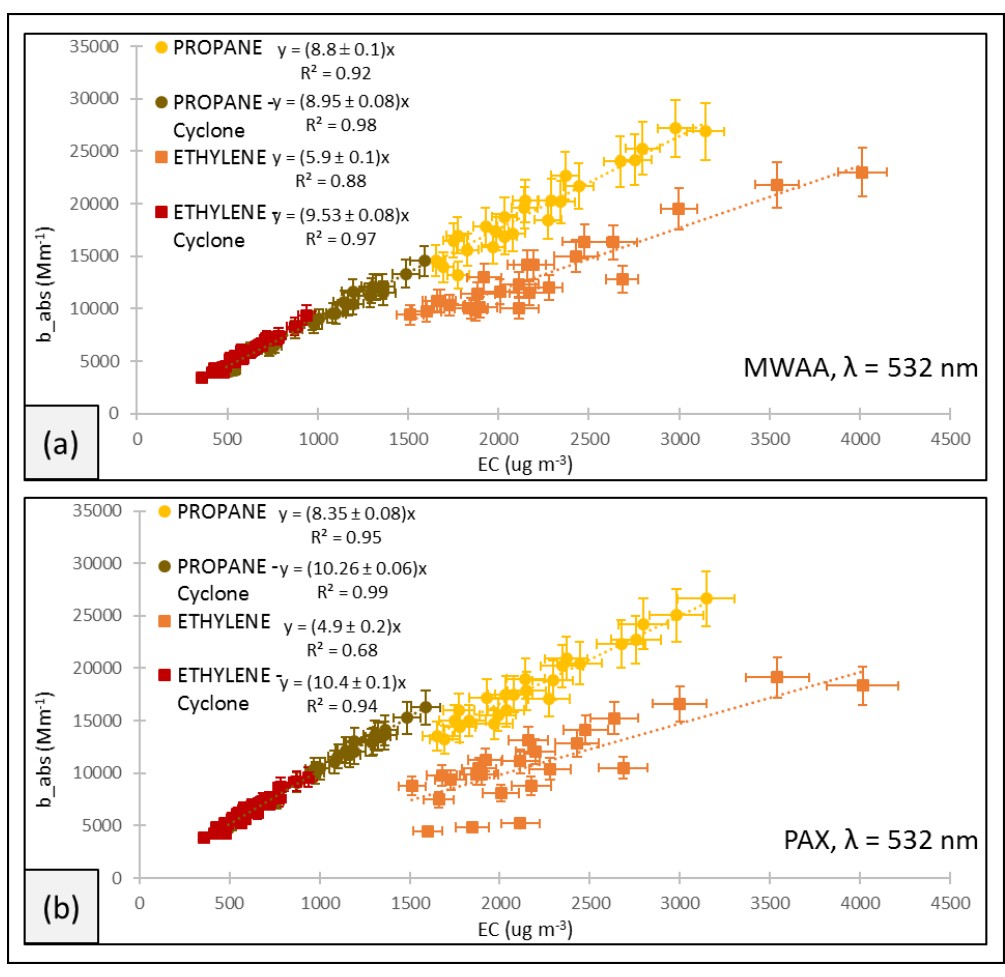


*Figure 13: Absorption coefficient @ 532 nm, measured by MWAA (a) and PAX (b versus EC concentration. The slope of*
*each fit corresponds to the Mass Absorption Coefficient.*

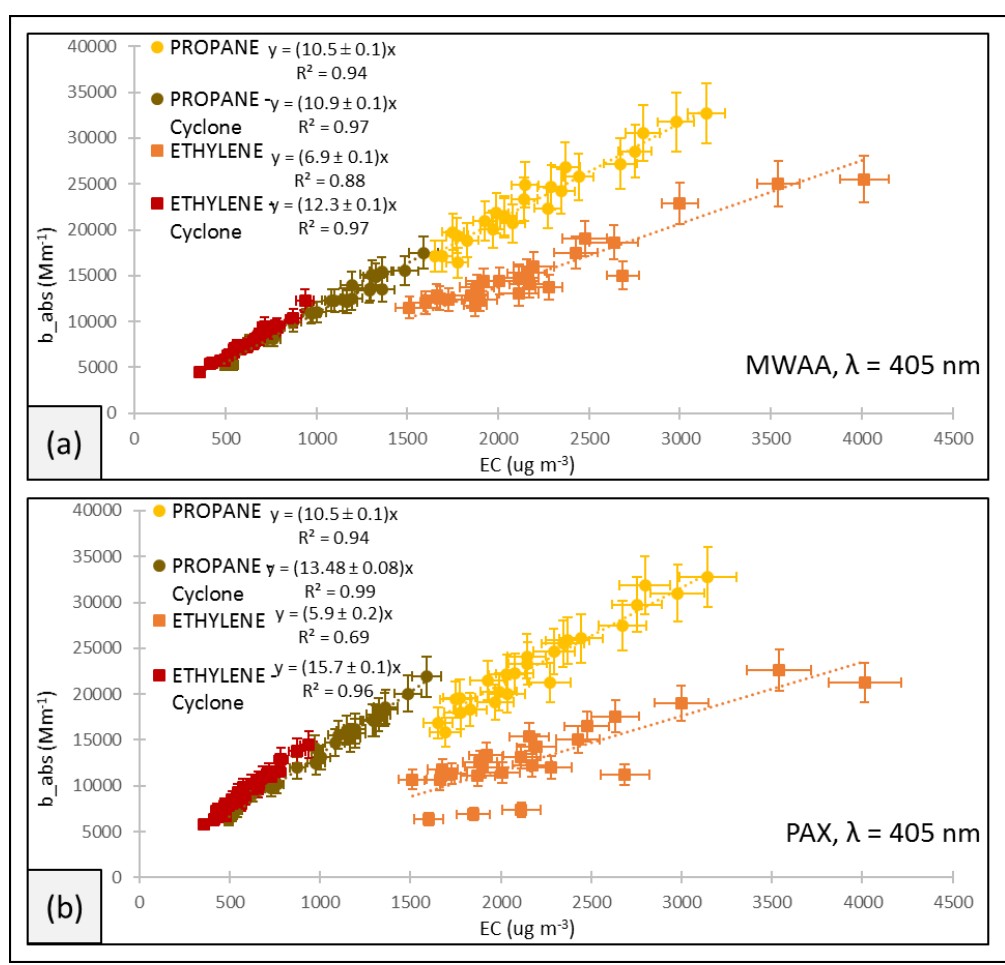

*Figure 14: Absorption coefficient at λ = 405 nm, measured by MWAA (a) and PAX (b) versus EC concentration. The slope of each fit corresponds to the Mass Absorption Coefficient.*

A summary of all the measured MAC values, including the other two wavelengths available for the MWAA (i.e., 635 and 375 nm) too, is given in Table 5. MAC values are close to theoretical figures for soot (Bond and Bergstrom, 2006), for both the fuels and at all the wavelengths. IR values are similar to those obtained by Moallemi (2019) for propane exhaust. With both the fuels MAC values increase when super-micrometric particles were removed by the cyclone; propane-particles showed higher MAC values than ethylene ones.





*Table 5: Summary of the measured MAC values, in m² g⁻¹.*

| FUEL | PAX | | | MWAA | | | | |
|---|---|---|---|---|---|---|---|---|
| | 870 nm | 532 nm | 405 nm | 850 nm | 635 nm | 532 nm | 405 nm | 375 nm |
| PROPANE | 5.30 ± 0.06 | 8.35 ± 0.08 | 10.5 ± 0.1 | 5.22 ± 0.06 | 7.22 ± 0.09 | 8.8 ± 0.1 | 10.5 ± 0.1 | 10.9 ± 0.1 |
| PROPANE with cyclone | 6.27 ± 0.06 | 10.26 ± 0.06 | 13.48 ± 0.08 | 5.32 ± 0.06 | 7.37 ± 0.07 | 8.95 ± 0.08 | 10.9 ± 0.1 | 11.6 ± 0.1 |
| ETHYLENE | 3.3 ± 0.1 | 4.9 ± 0.2 | 5.9 ± 0.2 | 3.78 ± 0.08 | 5.0 ± 0.1 | 5.9 ± 0.1 | 6.9 ± 0.1 | 7.3 ± 0.1 |
| ETHYLENE with cyclone | 5.41 ± 0.08 | 10.4 ± 0.1 | 15.7 ± 0.1 | 5.21 ± 0.06 | 7.62 ± 0.07 | 9.53 ± 0.08 | 12.3 ± 0.1 | 13.0 ± 0.1 |

In (Moallemi et al., 2019) only IR-MAC values are reported for the propane and they resulted slightly lower
than values here reported. This difference could depend on the different techniques used to quantify the EC
concentration: we used EC concentration from thermal optical analysis while Moallemi (2019) used BC
concentration measured by LII.
Discrepancies between MAC values obtained from PAXs and MWAA, for the same experiment, are
compatible with the differences of measured b_abs values. The b_abs values measured by PAXs and MWAA
are directly compared in Fig. 15, merging all the data collected by the two setups (i.e., with and without the
cyclone) and for the two fuels. The agreement between the two analyses turned out within 25 % and 7 %,
respectively without and with the cyclone.

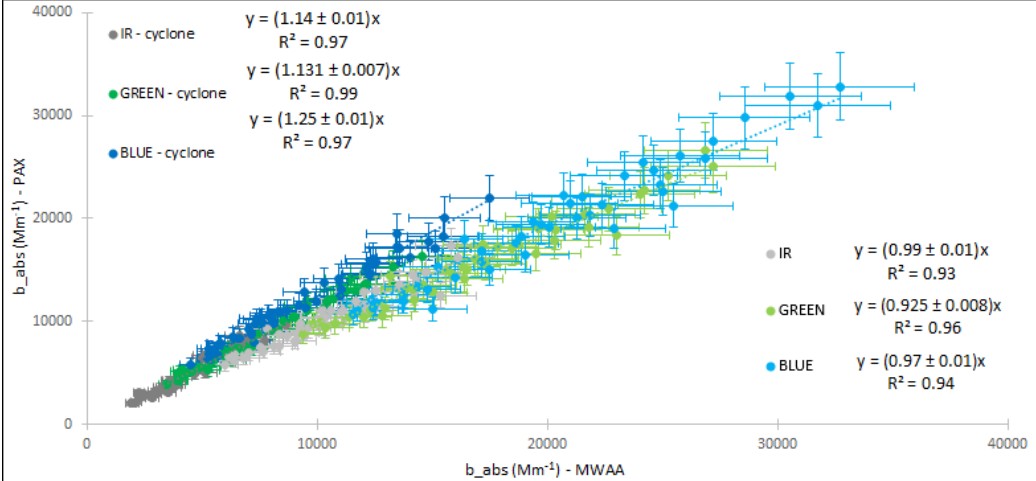


*Figure 15: Correlation study between the absorption coefficient measured by PAX and MWAA. Colours of identify the*
*wavelength of the analysis: grey refers to 870 nm, green to 532 nm and blue to 405 nm; dark and light colours refer to*
*experiments with and without cyclone, respectively.*
In addition, the spectral dependence of the absorption coefficient b_abs, and consequently the Ångström
Absorption Exponent (AAE, Moosmüller et al., 2011), can be calculated by the power-law:
$$b\_abs\ (\lambda) \approx \lambda^{-AAE}$$
where:
b_abs [Mm⁻¹]: absorption coefficient
$\lambda$ [nm]: wavelength used for the analysis
AAE: Ångström Absorption Exponent.



The averages of the resulting AAEs for the different experimental conditions are reported in Table 6.
Experimental determinations of the AAE had been reported in the literature as being dependent on aerosol
chemical composition (Kirchstetter et al., 2004; Utry et al., 2013) and size and morphology (Lewis et al., 2008;
Lack et al., 2012; Lack and Langridge, 2013; Filep et al., 2013; Utry et al., 2014 a). Particulate generated by
fossil fuel combustion (i.e., Black Carbon) typically has AAE values close to 1.0 (Harrison et al., 2013, and
references therein). The AAE values measured in this work for the MISG exhausts are generally close to 1.0
with higher figures for the cyclone-selected aerosol.

*Table 6: AAE values obtained in different experimental conditions through the analysis of PAXs MWAA raw data*

| EXPERIMENTAL CONDITIONS | AAE - PAX | AAE - MWAA |
|---|---|---|
| **PROPANE 70 to 85 mlpm - AIR 7 lpm** | 0.88 ± 0.06 | 0.92 ± 0.04 |
| **PROPANE 70 to 85 mlpm - AIR 8 lpm** | 0.92 ± 0.06 | 0.91 ± 0.05 |
| **PROPANE 70 to 85 mlpm - AIR 7 lpm - cyclone** | 0.98 ± 0.09 | 1.0 ± 0.1 |
| **PROPANE 70 to 85 mlpm - AIR 8 lpm - cyclone** | 1.05 ± 0.04 | 0.97 ± 0.09 |
| **ETHYLENE 118 to 144 mlpm - AIR 7 lpm** | 0.9 ± 0.3 | 0.84 ± 0.07 |
| **ETHYLENE 118 to 144 mlpm - AIR 8 lpm** | 0.76 ± 0.04 | 0.81 ± 0.06 |
| **ETHYLENE 118 to 144 mlpm - AIR 7 lpm - cyclone** | 1.40 ± 0.05 | 1.19 ± 0.09 |
| **ETHYLENE 118 to 144 mlpm - AIR 8 lpm - cyclone** | 1.39 ± 0.04 | 1.08 ± 0.05 |



**4. Conclusion**
A Mini-Inverted Soot Generator (MISG) was coupled with an atmospheric simulation chamber
(ChAMBRe) to compare the emissions of two fuels, ethylene, and propane. Different combustion conditions
(i.e., air and fuel flow, global equivalence ratio) were characterized in terms of size distribution, particle and
gas composition, optical properties, and EC concentration in the exhaust.
The MISG turned out to be a stable and reproducible soot particles source, suitable for experiments in
atmospheric simulation chambers. In addition, properties of emitted soot particles can be modulated by varying
the combustion conditions i.e., tuning the global equivalence ratio and/or varying the fuel used for combustion.
With equal conditions, ethylene combustion produces particles with higher number concentration and
smaller diameter than propane but is prone to generation of super-aggregates. These are likely formed directly
in the exhaust line where particles density is very high.
The carbonaceous compounds produced by propane are generally characterized by higher EC to TC ratios
than ethylene.
From the optical point of view, particles generated by propane turned out to be more light absorbing than
those formed by ethylene, although burning conditions (in terms of global equivalence ratio) were the same.
The values of the MAC parameter show a substantial agreement except those retrieved from the data collected
in the ethylene-no cyclone experiments. The latter resulted in lower MAC values, probably due to the presence
of super-aggregates in the chamber.
This work opens to new and more complex experiments. Well-characterized soot particles could be used to
investigate the effects that atmospheric parameters such as temperature and relative humidity can have on soot
particles, and also to study the interactions between soot particles and gaseous pollutants, solar radiation or
bio-aerosol.
**Author contribution**



VV and DM prepared the experimental setup, performed all the experiments and the data analysis; DM, FP,
and PP designed and built ChAMBRe; MB designed and implemented the acquisition software; VV and DM
prepared the article with contributions from the other authors.
**Competing interests**
The authors declare that they have no conflict of interest.
**Acknowledgments**
This project/work has received funding from the European Union's Horizon 2020 research and innovation
program through the EUROCHAMP-2020 Infrastructure Activity under grant agreement No 730997.

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
