# Peer review of "Characterization of soot produced by the Mini Inverted Soot Generator with an atmospheric simulation"

_Atmospheric Measurement Techniques, 2021_

## Author Comment (AC1)

Atmos. Meas. Tech. Discuss., referee comment RC1
https://doi.org/10.5194/amt-2021-345-RC1,   2021

[Figure]

**Comment on amt-2021-345**

Anonymous Referee #1
* * *
Referee comment on "Characterization of the MISG soot generator with an atmospheric simulation chamber" by Virginia Vernocchi et al., Atmos. Meas. Tech. Discuss., https://doi.org/10.5194/amt-2021-345-RC1, 2021
* * *
We thank the Referee for his valuable comments. In the following, we reply point-by-point to his notes.

This manuscript describes the properties of soot generated by a mini inverted soot generator in terms of optical and physical properties, such as particle size, EC/TC mass fraction and absorption coefficient. The soot particles were generated by combustion of propane and ethylene.

The paper is written in clear language and is technically sound. However, there have already been several papers published that characterize the MISG soot properties under various operating conditions (both with ethylene and propane combustion), and which are cited in this manuscript. The results of this paper are largely consistent with these prior papers (which is good), but the novelty of the results presented in this paper is not clear.

In the revised version we will highlight, as suggested as well by other Referees, the novelty of our work. We added in the text:

Line 13: This work deepens and expands the existing characterization of this soot generator that is also coupled with an atmospheric simulation chamber. Differently from previous works, MISG performance has been also tested at different fuel flows and higher global equivalence ratios. MISG exhausts were investigated after their injection inside the atmospheric simulation chamber: this is another novelty of this work.

Line 24: The soot characterization opens to various kinds of experiments in ASCs. Particles with well-known properties can be used, for example, to investigate the possible interactions between soot and other atmospheric pollutants, the effects of meteorological variables on soot properties and the oxidative and toxicological potential of soot particles.

Line 63: Differently from previous works (Bischof et al., 2019; Kazemimanesh et al., 2019; Moallemi et al., 2019), the MISG has been connected directly to an atmospheric simulation chamber; performance has been tested also at different fuel flows and higher global equivalence ratios. The present characterization deepens and expands the existing knowledge on particles and gases produced by this soot generator. The comprehensive characterization of the MISG soot particles is an important piece of information to design the subsequent experiments. Well-characterized soot particles could be used to investigate the effects that atmospheric parameters can have on soot particles, and also to study the interactions between soot particles and other pollutants.

No attempt was made to further develop the MISG or the atmospheric simulation chamber.

The MISG is a commercial instrument specifically designed for producing soot particles with very few margins of modification. Anyway, we think that any (possible) modification should come after a deep and exhaustive comprehension of how the MISG performs in terms of soot production and properties. The modification/development of an atmospheric simulation chamber is beyond the scope of this paper, even if for sure interesting in this field of research.

Moreover, the link to atmospheric sciences is missing (for instance, an intercomparison of aerosol measurement instruments using MISG soot as test aerosol, validation of new measurement techniques or data analysis procedures etc.). In this sense, I believe that this manuscript does not fit so well into the scope of AMT.

As the Referee underlines, soot generators can be useful in several ways to study atmospheric processes and test related instrumentation. Therefore, we decided to investigate the MISG performance in detail. Moreover, the manuscript has been submitted to the AMT special issue "Simulation chambers as tools in atmospheric research". We consider the focus of our work coherent with the special issue scope: differently from previous studies, we characterized soot particles inside the atmospheric simulation chamber and not directly at the MISG outlet.

I would suggest to streamline the discussion on the soot characterisation (some of the results can be shifted to Supplemental Information) and add new results related to atmospheric sciences (e.g. interactions of soot with gaseous pollutants and bio-aerosols as suggested in Section 4 "Conclusions"). This would enhance the novelty of the paper. Another option would be to submit the manuscript to another journal, which is focused more on laboratory instrumentation.

Our paper focuses on the characterization of soot particles inside a simulation chamber. The interaction between soot particles and bio-aerosol, as well as the interaction with the gas phase are the next steps of our research, but they are beyond the scope of the present work.

Technical comments:

Line 46: The authors state that "The Inverted-Flame Burner (Stipe et al. 2005) is often considered as an ideal soot source (Moallemi et al., 2019 and references therein), due to its capacity to generate almost pure-EC particles and for the stability of the flame and of its exhaust (Stipe et al. 2005). To such category belongs the Mini-Inverted Soot Generator (MISG) used in this work". I find this sentence somewhat misleading. The MISG is known to suffer from poor day-to-day reproducubility for particle sizes below 150 nm (Moallemi et al.).

In the dedicate paragraph (§3.3) Moallemi et al. reported the repeatability (R) of the MISG in terms of 1) mode diameter, 2) number concentration, 3) mass concentration. In all the three cases they found that R depends mainly on the air flow given to the soot generator: in general, the lower is the air flow, the better is R. What is actually demonstrated by Moallemi et al., is that the MISG repeatability is very dependent on the feeding flows: it appears to be dependent on the global equivalence ratio and therefore on the flame shape. Indeed, it is shown that the variability increases (Fig. 5) when the combustion conditions are near to the Flickering flame operative range (Fig. 3).
By carefully choosing the pair of air and fuel flow values entering the MISG, the repeatability is more than satisfactory for all the three parameters. About the repeatability for particles below 140 nm, Moallemi et al. reported a poor day-to-day repeatability without defining a numerical range. It is worthy to note that the Authors underline how these values are affected not only by the burner, but also by the dilution system stability and the instruments measuring particles properties.

This might be the reason why in the current manuscript the authors only generated soot with mode diameters larger than 170 nm (Figure 4). Considering that most engines emit ultrafine soot particles (with GMDmob from about 20 nm for aircraft engines up to about 90 nm for diesel vehicles), it is not clear to me how the MISG can generate realistic soot size distributions.

We agree with the referee about the capability of this kind of burners to produce ultrafine soot particles: the size distribution also inside the chamber is picked to bigger particles while real engines produce smaller ones. However, our statement is not about engines, but limited to ideal soot sources. If necessary, we will specify in the text that this generator is not suitable for ultrafine particles.

Particles with mobility diameter <100 nm could potentially be size selected, e.g. with a DMA, but this was not investigated in this study. Moreover, it is not clear whether the number concentration of the size-selected particles (after being diluted in the chamber) would be sufficiently high for most type of experiments.

The possibility to use a DMA to inject inside the chamber ultrafine particles only is reasonable and can be investigated. We think that concentrations useful for most kind of experiments can be reached easily just increasing the injection time favouring the accumulation inside the chamber.

---

## Author Comment (AC2)

Atmos. Meas. Tech. Discuss., referee comment RC5
https://doi.org/10.5194/amt-2021-345-RC5, 2021

[Figure]

**Comment on amt-2021-345**

Anonymous Referee #4
* * *
Referee comment on "Characterization of the MISG soot generator with an atmospheric simulation chamber" by Virginia Vernocchi et al., Atmos. Meas. Tech. Discuss., https://doi.org/10.5194/amt-2021-345-RC5, 2021
* * *
**Review of "Characterization of the MISG soot generator with an atmospheric simulation chamber"**

We thank the Referee for his valuable comments. In the following, we reply point-by-point to his notes.

**General comments:**

This paper discusses the physical, chemical, and optical properties of soot produced by burning propane and ethylene in a miniature inverted soot generator (MISG). Although some aspects of this work (such as flame shape vs. fuel and air flows) have been discussed in previous studies, there are some novel aspects to the paper: combining the MISG with an atmospheric simulation chamber and studying the optical properties of soot in depth. The methodology used in the paper is sound and valid but the paper itself is cluttered and poor in terms of readability. I suggest that the authors streamline the paper by omitting the discussion on flame shape with combustion conditions or moving it to the supplementary material, and instead focus on aspects that have not been covered in similar other studies.

We have followed the Referee suggestion, moving tables 3 and 4 to Supplementary and improved the general quality of the text.

The whole manuscript should be edited for grammar and proper academic writing too. There are also some discrepancies between the results presented in this paper and previous papers that characterized the MISG soot, which need to be discussed in more details by the authors (see my comments below). Overall, the paper is not acceptable in its current form and needs major revisions before it can be published.

We improved the text as suggested.

**Specific comments:**

Article title: Avoid using an acronym in the title without fully defining it first.

Done. We have modified the title following both RC2 and RC5 comments.

The revised title is: "Characterization of soot produced by the Mini Inverted Soot Generator with an atmospheric simulation chamber"

Abstract should be written as one paragraph.

Done.

Abstract: MAC stands for mass absorption cross-section, not mass absorption coefficient.

Done.

Section 1: Combine paragraphs 2 and 3.

Done.

Line 45: List the "several other purposes" in those references more specifically.

Done. In the revised version we added:

Line 45: "such as studies on atmospheric processing of soot particles, characterization of uncoated/coated and fresh/denuded of soot particles"

Section 1: The introduction is written poorly and needs to be improved in terms of readability and transition between paragraphs. It should also clearly state the objective(s) and novelty of the study near the end of the introduction.

We added some information and clarified better the purpose of the work:

Line 13: This work deepens and expands the existing characterization of this soot generator that is also coupled with an atmospheric simulation chamber. Differently from previous works, MISG performance has been also tested at different fuel flows and higher global equivalence ratios. MISG exhausts were investigated after their injection inside the atmospheric simulation chamber: this is another novelty of this work.

Line 63: Differently from previous works (Bischof et al., 2019; Kazemimanesh et al., 2019; Moallemi et al., 2019), the MISG has been connected directly to an atmospheric simulation chamber; performance has been tested also at different fuel flows and higher global equivalence ratios. The present characterization deepens and expands the existing knowledge on particles and gases produced by this soot generator. The comprehensive characterization of the MISG soot particles is an important piece of information to design the subsequent experiments. Well-characterized soot particles could be used to investigate the effects that atmospheric parameters can have on soot particles, and also to study the interactions between soot particles and other pollutants.

Line 69: Change "air and fuel flow in an opposite way to the buoyancy force" to "… in opposite direction to the …"

Done.

Line 70: Change to "The resulting diffusion flame is more stable by reduced flickering of flame tip"

Done.

Line 77: lpm and mlpm should be defined (it is better to use L/min or mL/min as units of flow rate).

Done.

Line 79: This statement is not correct. Kazemimanesh et al. (2019) states that part of the air flow is used in combustion and the rest is used to dilute the exhaust products.

Corrected.

Lines 81-101: The definition of equivalence ratio is based on fuel-to-air ratio, thus the reader would not be confused if you define the fuel-to-air ratio (instead of AFR) first. Also, all equations should be numbered.

Corrected.

Line 93: The units used for AFR are not clear to me. AFR is a unitless parameter, so just get rid of any units.

Done.

Line 103: Many of the in-text citations in this article should be in format of Author (Date). Please consider this whenever suitable during revision. For example: Moore et al. (2014) demonstrated that fuel-lean flames produce soot particles …

For in-text citations, we followed the AMT guidelines that say "If the author's name is part of the sentence structure only the year is put in parentheses ("As we can see in the work of Smith (2009) the precipitation has increased"). If the author's name is not part of the sentence, name and year are put in parentheses ("Precipitation increase was observed (Smith, 2009)")"

Line 144: Consider changing to "at the fuel tube nozzle"

Done.

Line 125: I cannot find Section 2.1.2 in the paper.

This section was not present and has been deleted. We thank the Referee for noticing this material mistake.

Line 179: It is known that the multiple charge correction algorithm in the TSI AIM software breaks when the median mobility diameter is relatively large (>200 nm). Can the authors show the uncorrected and corrected size distributions for 2-3 cases in the supplementary material?

While checking the figures, we noticed that data were not corrected for the multiple charge correction algorithm. We apologize for this mistake in our text. We added a pair of uncorrected and corrected size distributions in the supplementary material (Fig. S1).

Line 218: Change peculiar to a better adjective.

Done.

Line 236: "To our knowledge, no literature information is available for the ethylene in the flow range of Table 4." This statement is not true. Kazemimanesh et al. (2019) studied the MISG and its flame shape with ethylene and air flow rates (80-130 mL/min and 4.0-10.0 L/min, respectively) that partly cover Table 4.

Modified in "A similar characterization with ethylene also exists but it only partly covers the flow ranges explored in the present work. We got some differences especially in the transition range to Open tip flames, probably due to the different setups. Also the subjectivity of the visual determination, that is user dependent, can lead to differences."

Lines 241-251: The authors talk about various experiments that they did and the calculated repeatability (mistakenly noted as "reproducibility") in mode diameter and concentration. However, it is not clear what conditions were tested and the results are not shown in the paper or the supplementary material.

Corrected and specified. The conditions tested are reported in Table 1 and 2, while results are discussed in the text.

Page 10 – Fig. 4 and the discussion around it: The particle mode diameter reported for ethylene flames is constant at ~175 nm. This is inconsistent with previously reported values of ~240 nm and up to 270 nm (Kazemimanesh et al., 2019). The same reference also reported an initial sharp increase in particle size and concentration with increasing ethylene

flow rate, which eventually levelled off to a relatively constant value. This is in contrast to the trend seen in this paper. These differences must be noted and discussed in the paper.

We added the discussion about these differences, that probably depended on the different combustion conditions.

Line 292: Even if the direct comparison between our findings and results from previous works (Bischof et al., 2019; Kazemimanesh et al., 2019; and Moallemi et al., 2019) are not directly comparable (since feeding flows and global equivalence ratios are different), some similarities can be identified. Previous works observed that by increasing the fuel flow, the particle number concentration increases too, that is what we observed for propane. In addition, Bischof (2019) also reported that the particle mode diameter, with propane, did not depend on the global equivalence ratio, as we also observed, but for ethylene. Kazemimanesh (2019) showed a clear increase in mode diameter, corresponding to an increase of fuel flow rate, that reached a quite constant value (i.e., around 240-270 nm) for ethylene. This trend differs from our observations, since the mode diameter in our case turned out to be quite stable at about 175 nm independently on feeding flows. This difference is probably due to the global equivalence ratios used: while in (Kazemimanesh et al., 2019) global equivalence ratios are lower than 0.206, in our case they are higher than 0.213. In (Moallemi et al., 2019), instead, they observed an opposite behaviour for mode diameters: they retrieved that at fixed fuel flow, a higher air flow produced a slight decrease of the mode diameter. Both (Moallemi et al., 2019) and (Bischof et al., 2019) measured mode diameters < 200 nm, but they used different combustion conditions (i.e., lower global equivalence ratios resulting from higher air flow or lower fuel flow). We can conclude that, as expected, global equivalence ratio is the principal parameter affecting size distributions of soot particles.

Anyway, as request by RC2, we will perform experiments that replicate the conditions used in the previous works, so we will able to compare the same operative conditions used by (Kazemimanesh et al., 2019).

Fig. 3 and 4: The authors should consider adding error bars to the data points. In addition, it is not clear why a linear fit is shown for the data points when the paper does not offer any evidence or support for trend.

Done and specified ("Lines aim to facilitate the reader eye."). The same changes were applied to Fig. 7, 8, 9 and 10 too.

Fig. 5: I suggest that the authors show and discuss figure 5 before figures 3 and 4, as this will enhance the readability and flow of the paper. I was completely lost about the results shown in figures 3 and 4 when I first read the paper until I saw figure 5. Figures 3 and 4 are essentially the size distribution parameters extracted from figure 5 and shown with respect to equivalence ratio.

Modified as suggested.

Lines 307-316: Can you show the number and volume distributions side by side in Fig. 6? What is meant by "relative particle number concentration" in Fig. 6? [dN/dlog dp]/N_tot?

We noticed some inaccuracies in the text about the kind of discussed distributions and we corrected them. Since in the text we discussed number and mass distribution, we modified Fig.6 by showing number and mass distribution side by side. In addition, the caption was modified by changing "relative concentration" to "normalized concentration"

Section 3.2.3 (EC-OC analysis): The authors did not elaborate how they calculated TC (total carbon). OC can exist in gas-phase or as condensed semi-volatile particles and the authors need to distinguish between the two when calculating TC. The authors briefly mention the use of a second filter, which should help in determining the mass concentration of OC existing as semi-volatile particles.

Total Carbon was calculated by the thermal-optical analysis as the sum of the whole evolved carbon during the analysis. The instrument was calibrated by using a standard solution.
We used backup filters to estimate the semi volatile/volatile fraction of OC, which resulted compatible with the organic contamination on blank filters. Hence, the total OC is given by the concentration value measured on the main filters. We are not able to evaluate OC in gas-phase.

Fig. 9 and 10: I do not quite understand why normalized EC concentrations are shown rather than the absolute values of EC concentration or the EC/TC ratio. The latter two parameters are more important for researchers when using a soot generator.

We have shown the normalized concentrations to emphasize the differences deriving from the use of the cyclone in case the soot is produced by the combustion of propane or ethylene. Anyway, the EC values are those measured on the filter and deriving from the concentration in the chamber and not directly produced by the soot generator.

Line 384-385: Why is propane soot more light absorbing than ethylene soot at all three wavelengths?

This is a very good question. Optical properties such as absorption depend on several parameters, mainly composition, mixing state, aging, and size. Considering all the experiments reported in this work, no differences in composition can be expected, since only EC particles are present: this means that differences in absorption cannot depend on particle composition. Also mixing state and aging can not explain this difference: soot inside the chamber is fresh and only EC is present. We can explain the higher light absorbing capability of propane by considering differences in: size distributions (see Figs. 3-5) and morphology/density of the particle produced by the burning of the two different fuels. We have added in the revised text these considerations.

Line 498: The formation of superaggregates is related to high particle concentration in the exhaust line. This means that by diluting the MISG exhaust, the formation of these large aggregates can be alleviated. Kazemimanesh et al. (2019) and Chakrabarty et al. (2012) suggest that these superaggregates are formed at the stagnation plane of the flame tip, which seems more plausible. The authors should note and discuss these differences in the paper (not in the conclusions section).

This point has been raised by more than one Referee, and we agree that it is an interesting point to investigate. We will be able to answer to this question and add the results in the revised text after some extra experiments, by inserting a diluter between MISG and ChAMBRe as suggested. We will also try to modify the injection line length, as suggested by RC2. Since our atmospheric chamber is currently engaged full time in non-postponable experiments, we will perform the experiment as soon as possible, in agreement with the editor.

---

## Author Comment (AC3)

Atmos. Meas. Tech. Discuss., referee comment RC3
https://doi.org/10.5194/amt-2021-345-RC3,   2021

[Figure]

**Comment on amt-2021-345**

Anonymous Referee #3
* * *
Referee comment on "Characterization of the MISG soot generator with an atmospheric simulation chamber" by Virginia Vernocchi et al., Atmos. Meas. Tech. Discuss., https://doi.org/10.5194/amt-2021-345-RC3, 2021
* * *
OVERVIEW

The present manuscript aims to characterize soot particles produced with a MISG. Overall the manuscript needs some rewriting and rethinking. Many sections appear to be a list of results which are scarcely interpreted, while many figures are barely described. Hence, some important results are shaded by many non-relevant information and figures. As final results, the conclusions and overall take-home message of the manuscript becomes extremely unclear. As mentioned in the second review, the authors show for the first time that MISG can produce large soot aggregates, which are not well characterized in other previous studies and cannot be generated, up to my knowledge, with the more traditional CASTburner. So, I suggest the authors to focus on this specific aspect of their research. Considering the presence of these large particles, PAX measurements must be corrected for truncation error. In its current form, the manuscript is not suitable for publication.
* * *
We have taken seriously the Referee's general comments, which we thank for the useful observations and suggestions for improvement. We have revised the manuscript, trying to put in evidence the new findings, and improving the conclusions with a more clear take-home message. In the following, we reply point-by-point to his notes.

SPECIFIC COMMENTS

ABSTRACT: The abstract is very generic and does not provide any real information on the performances of the burner. I suggest rewriting of the abstract.

Done. In the revised version, we added:

Line18:  Significant differences could be observed when the MISG is fuelled with ethylene and propane both in terms of particle size, in particular, the production of sub-micrometric super aggregates was observed for ethylene combustion. With equal combustion conditions, ethylene produced higher number concentration of particles and smaller mode diameters. Soot particles produced by propane combustion resulted in higher EC:TC ratios and they were more light absorbing than particles generated by ethylene combustion.

INTRODUCTION: Introduction does not provide a context and does not present the motivation for this study. At the moment is a list of references without a clear story behind it. It does need some rewriting.

We added some information about the context and motivation of this study.

Line 39: BC is considered one of the most significant radiative forcing agent, second only to $CO_2$ (Ramanathan and Carmichael, 2008; Bond et al., 2013). Another positive effect on radiative forcing is related to the darkening of glaciers surface due to the deposition of BC (Skiles et al., 2018). Soot contributes to air pollution also via reactions with several gas species, as $NO_2$, $SO_2$ and $O_3$ (Finlayson-Pitts and Pitts, 2000; Nienow and Roberts, 2006).

Line 40: Soot particles are suspected to be particularly hazardous to human health, because they are sufficiently small to penetrate the membranes of the respiratory tract and enter the blood circulation or be transported along olfactory nerves into the brain (Nemmar et al., 2002; Oberdörster et al., 2005).

Line 42: In this context, soot generators are employed as stable sources of soot particles.

Line 59: ASC experiments are the best compromise between laboratory and field experiments, since they simulate real situations but without the uncertaintes and variability of typical field measurements.

Line 63: Differently from previous works (Bischof et al., 2019; Kazemimanesh et al., 2019; Moallemi et al., 2019), the MISG has been connected directly to an atmospheric simulation chamber; performance has been tested also at different fuel flows and higher global equivalence ratios. The present characterization deepens and expands the existing knowledge on particles and gases produced by this soot generator. The comprehensive characterization of the MISG soot particles is an important piece of information to design the subsequent experiments. Well-characterized soot particles could be used to investigate the effects that atmospheric parameters can have on soot particles, and also to study the interactions between soot particles and other pollutants.

L27-34: the definition of "BC" is mostly based on its non-null imaginary part of the refractive index. However, Petzold 2013 made it clear that the term BC is a qualitative definition of BC rather than operational. From your text I have the impression that absorption photometers will directly provide BC concentration. I think that, however, a discussion on soot nomenclature is not needed so early.

We are aware of the paper by Petzold 2013. Our incipit was just to underline that BC and EC quantities depend on the measuring technique used to determine them. In this sense, these quantities are operationally defined. We consider these first lines (and the references therein) as a very brief introduction to the topic which could be useful for some readers.

L43-50: This part needs to be developed further since it will create the right context for your work.

We thank the Referee for the valuable suggestion. We have extended this part to better introduce our work. In the revised version we added:

Line 45: such as studies on atmospheric processing of soot particles, characterization of uncoated/coated and fresh/denuded of soot particles

L51-63: This part provides some sparse technical details of burner and a very generic description of smog chambers. It is not clear what the author wants to communicate here.

In this part, we establish the link between the main components used in our work (i.e. soot generator and ASC) before entering in the material and methods section. We would like to keep this part unmodified.

L74: reference

Done.

L77-80: what are the consequences of the absence of quenching or carrier gas?

Actually, the quenching or carrier gas are present in the MISG too, since a fraction of the feeding air is not used for combustion but used as quenching/carrier of the output flow. The differences between the miniCAST and the MISG are basically two: with the miniCAST, the quenching gas is N2 instead of air, so the quenching effect is reasonably higher than in the case of air. Second, in the case of the miniCAST the quenching gas flow can be selected, thus modifying the flame shape and flow turbulence. In the MISG, the quenching gas is air and the flow can not be selected, since it is just a fixed fraction of the feeding flow.

L84: give number to equation. Recurrent

Done.

L85-91: this occupy more space than needed. Put it as normal text. Recurrent

Done.

155-156: revise indent.

Done.

L166: size distribution measurements

Done.

L181-182: what refractive index was used to derive diameter from OPS?

The default OPS setting was used, i.e. 1.59.

L196-197: Considering the extensive use of PAX measurements in the paper I am genuinely surprised that truncation errors are disregarded. I think it is important to show that truncation is not relevant in these conditions. As recently resumed by Modini et al. (2021) little is known on the dependency of scattering phase function on the particle morphology and how this might impact truncation for highly absorbing aerosol particles. Scattering correction for absorbing aerosol is investigated for the nephelometer instrument by Bond et al., 2009. The argument of the authors is valid, but it should be contextualized if not verified.

We agree with the Referee regarding the opportunity to correct PAX data for truncation errors. By the way, the Modini et al. paper shows how the truncation error can be non-negligible on real aerosol samples when the SSA values are above 0.85. In our work, all the aerosols produced in the chamber have SSA value below 0.3, not surprising considering that they are composed by pure fresh BC particles. Moreover, in the paper by Modini et al., they used a CAPSssa instrument, which principle of operation is completely different from PAXs. It is not clear to us how the truncation error could affect the response of the microphone integrated in the PAX, but in principle we don't expect a significant bias. We didn't know the paper from Bond et al., 2009 and we thank the Referee for bring it to our attention.

Following the suggestion given by the Referee, we will add in the revised text the following sentence

Line 195: "Few papers in literature deal with the correction for truncation errors in nephelometer measurements (Bond et al., 2009, Modini er al, 2021) for highly absorbing particles: little is known on the dependency of scattering phase function on the particle morphology and how this might impact truncation.".

L246-251: If I understand correctly this is simply the relative standard deviation. It is not clear, however, in what conditions these values were calculated.

Yes, it is the relative standard deviation; we added the definition in the text. It was calculated by performing the same experiment many times. This is true for all the combustion conditions listed in Table 1 and 2.

L301-302: does it mean that all size distribution are measured 3 minutes after injection.

It means that the soot injection from the MISG into the chamber lasted 3 minutes and the measurements started just after the mixing time (other 3 minutes). We specified this information in the revised text.

The ageing time in the chamber should be always specified, since concentration and dimeter of particles drastically change due to coagulation, especially at high concentrations.

Specified in Sect. 3.2.1. We added:

Line 302: Data were acquired starting 3 minutes (i.e., the chamber mixing time) after the MISG switching off, for a specific time interval (i.e., 4 to 10 minutes). All the curves are normalized to the same injection time (i.e., 3 min of injection inside the chamber).

L340-372: Describing both EC:TC and OC:EC is redundant. It is also hard to compare and understand the impact of large soot on the OC:EC fraction from the two figures. I would suggest to merge them or focus on the impact of large soot. To be honest, figure 9 could easily go in the supplementary. Is there any correlation between OC:EC and diameter mode, CO2, NO ?

We changed the discussion from OC:EC to OC:TC and merged Fig.9 and 10. These figures emphasize the differences deriving from the use of the cyclone in case the soot is produced by the combustion of propane or ethylene. No correlations were observed between OC:EC and other parameters.

Section 3.2.4 -3.2.5: These sections could be merged. Especially considering the length of Section 3.2.4.

Done.

Figure 11 is barely described or discussed in the text. Hence, it can be moved to supplementary or removed.

We moved it to supplementary.

L412-417: This part is very hard to read and follow. The authors are requested to build a discussion on their result, rather than list numbers in series. This problem appears in almost every section of the paper. As a consequence, Figure 12-13-14 become hard to interpret too. I would suggest move the figures to supplementary, summarise the result in a table and construct a separate discussion for ethylene and propane.

We moved Fig.13 and 14 to the supplementary and Table 5 before Figure 12. We have re-written the discussion.

Line 412: The MWAA analysis at λ = 870 nm (Fig. 10.a) returned compatible MAC values for both propane series (with/without cyclone) and ethylene series with cyclone, while a consistently lower MAC value was found for the ethylene series (worse correlation) without the PM1 cutting. The same picture turned out at the other two wavelengths (in supplementary). By comparing PAX absorption coefficients and EC concentrations at λ = 870 nm (Fig. 10.b), obtained MAC values are more variable with similar values only in the case of propane without cyclone and ethylene with cyclone.  At λ = 532, in the case of MWAA, similar MAC values have been found for both the propane series, while, for ethylene series, MAC values were slightly higher when cyclone was used and lower when not. Considering the optical data from PAX, a similar MAC for both fuels was found when the cyclone was present, while it slightly differed in the case of propane without cyclone, and it was much lower in the case of ethylene without cyclone. At λ = 405 nm, the MWAA responses for propane series were still in agreement while the ethylene series showed a higher MAC value when using the cyclone, and a lower MAC value without using it. PAX returned a different MAC value for each of the four conditions. To summarize, if only series with cyclone are considered, MAC values show only small differences depending on the

fuel,larger in the case of PAXs. The ethylene series without cyclone showed the lowest MAC values of the whole data-set: the most likely reason for this difference is the presence of super-micrometric particles (see Sect 3.2.1 and Fig. 6) when the cyclone was not used. With MWAA, the MAC values turned out to be the same in all the runs but the case of the ethylene data collected without the cyclone. With the PAXs analysis, MAC values turned out higher in the series with cyclone, this happened at all the three wavelengths and for both fuels. Since PAXs data showed a higher variability in MAC values, photoacoustic measurements are supposed to be more sensitive to particle size than filter based MWAA analysis.

Figure 15: since you do not correct for truncation error, this comparison between PAX and MWAA is highly questionable, especially for the experiments without cyclone.

As discussed above, we think that the truncation error is not an issue in the case of photoacoustic measurements. This is true especially in the case of experiments with cyclone, where particles are smaller than 1 micron.

---

## Author Comment (AC4)

Atmos. Meas. Tech. Discuss., referee comment RC2 https://doi.org/10.5194/amt-2021-345-RC2, 2021 © Author(s) 2021. This work is distributed under the Creative Commons Attribution 4.0 License.

**Review of Vernocchi et al. MISG characterization**

Anonymous Referee #2

Referee comment on "Characterization of the MISG soot generator with an atmospheric simulation chamber" by Virginia Vernocchi et al., Atmos. Meas. Tech. Discuss., https://doi.org/10.5194/amt-2021-345-RC2, 2021

Review of 10.5194/amt-2021-345, Characterization of the MISG soot generator with an atmospheric simulation chamber, by Vernocchi et al.

We thank the Referee for his valuable comments. In the following, we reply point-by-point to his notes.

The authors characterized an Argonaut MISG (Model MISG-2) using measurements of size distribution, elemental carbon, and light absorption. These measurements are thoroughly described. Measurements were repeated for a number of fuel/air flows using ethylene and propane, and also with/without a cyclone to remove large particles. Size distributions were measured using mobility measurements (SMPS; up to about 800nm) and an optical particle sizer (OPS; up to about 8 um). The results are presented clearly and the authors have observed an important supermicron soot mode that has not been reported by the 3 previously published literature studies on the MISG-2, which is by itself a strong reason for publication.

The title and abstract should be modified to emphasize that conclusion, but at the same time the abstract must also mention that this study uses different fuel flow rates than earlier studies. A new title might be "Characterization of supermicron and submicron soot produced by a miniature-inverted soot generator". (Current title uses an acronym with 2 words which are part of the acronym.)

We modified the title and abstract as suggested.

The revised title is: Characterization of soot produced by the Mini Inverted Soot Generator with an atmospheric simulation chamber

We added in Line 13: "This work deepens and expands the existing characterization of this soot generator that is also coupled with an atmospheric simulation chamber. Differently from previous works, MISG performance has been also tested at different fuel flows and higher global equivalence ratios. MISG exhausts were investigated after their injection inside the atmospheric simulation chamber: this is another novelty of this work".

Line 24: The soot characterization opens to various kinds of experiments in ASCs. Particles with well-known properties can be used, for example, to investigate the possible interactions between soot and other atmospheric pollutants, the effects of meteorological variables on soot properties and the oxidative and toxicological potential of soot particles.

We have also added some statements about the scope and novelty of our work in the last part of the introduction.

Line 63: "Differently from previous works (Bischof et al., 2019; Kazemimanesh et al., 2019; Moallemi et al., 2019), the MISG has been connected directly to an atmospheric simulation chamber; performance has been tested also at different fuel flows and higher global equivalence ratios. The present characterization deepens and expands the existing knowledge on particles and gases produced by this soot generator. The comprehensive characterization of the MISG soot particles is an important piece of information to design the subsequent experiments. Well-characterized soot particles could be used to investigate the effects that atmospheric parameters can have on soot particles, and also to study the interactions between soot particles and other pollutants".

I have a few major comments which should be addressed before publication.

Major comments ------

The observation of supermicron soot would suggest that future studies should never use ethylene fuel in the MISG (and perhaps also other inverted burners) as a surrogate for atmospheric soot. This is an important conclusion, and although an earlier study using ethylene in the MISG (Kazemimanesh et al., 2019) noted the supermicron soot, its importance was not emphasized.

In the revised text, we will emphasize this issue in the abstract as well as in the conclusions.

That study also used a different flow rate. Given this emphasis I would like to request one additional experiment is made before publication. The authors should directly test their hypothesis that "super-aggregates...are likely formed directly in the exhaust line where particles density is very high" (lines 498-499). If this is the case, then could the issue be solved simply by diluting immediately after the MISG? The experiment would be simple. The authors need only to run the MISG with 3 line lengths. Very short, normal (as used previously), and very long. For each line length, measure with the OPS and SMPS. The results should be reported as combined OPS-SMPS size distributions in mass and number weighting.

This point has been raised by more than one Referee, and we agree that it is an interesting point to investigate. We will be able to answer to this question after some extra experiments, 1) by modifying the line length as suggested, 2) by inserting a dilution system just after the SG exhaust. Anyway, we have just a doubt about the effect produced by the modification of line length. After the small quartz cell where the flame burns, the exhaust is carried outside the SG after passing through a copper serpentine, with length roughly 40 cm long. If coagulation happens in this section, no way to understand if super-aggregates are formed in the flame or after.

Since our atmospheric chamber is currently engaged full time in non-postponable experiments, we will perform the experiment as soon as possible, in agreement with the editor.

Second, and continuing from above, a discussion of the physical properties of the supermicron aggregates is missing. For example, if super-aggregates are formed in the exhaust lines, then they should have the same MAC as the particles they are formed from.

If they do not, then they must have a different morphology. Chakrabarty et al.

(https://doi.org/10.1038/srep05508, 2014, Fig S3) predict a similar MAC for supermicron aggregates as for smaller aggregates. So, the authors might be observing aggregates that are more compacted than expected. This is supported by the trends in Figures 9 and 10. The authors should discuss their data with reference to this and other literature on superaggregates.

We thank the Referee for the interesting paper; however, it investigated soot emission by wildfires, not by laboratory sources. Moreover, we are confused about the indication of Figs. 9 and 10 about the interpretation of global equivalence ratios vs. EC concentration. Why should these Figs suggest that observed aggregates are more compacted than expected? About S3 figure by Chakrabarty et al.: it seems to us that the MAC is quite far from constant with respect to the size of the super-aggregates. If we interpret correctly this figure, the MAC is constant in respect to the single monomer.

We agree with the Referee that a full characterization of super-aggregates, independently if they are formed directly at the flame or after in the exhaust line, would be of great interest. At the same time, we think that such deep characterization is outside the scope and the possibilities of the present work. Anyway, we will try to answer to this point with the experiments discussed in the addendum.

Third, I would also request that the authors try harder to reproduce exactly the conditions used in previous studies. As it is, the authors have used higher fuel flow rates (equivalence ratios) than all previous studies (Kazemimanesh et al., 2019; Moallemi et al, 2019; Bischof et al., 2019). It is unclear to me why the authors have not reproduced previous measurements exactly, to allow for comparable results. Is it because the authors used long line lengths and changed the pressure downstream of the flame? Is it because the authors used an "MISG-2" and not an "MISG-1"? Also, as mentioned above, the abstract should emphasize this difference in flow rate.

The opportunity to reproduce exactly the condition used in previous works is for sure a good scientific procedure. However, we did not reproduce previous measurements for several reasons. First, our intention was to explore new operation conditions in order to expand the knowledge of the SG. Secondary, we wanted to compare the soot produced by propane and ethylene, since all the previous works focused on one fuel only at a time. So, we considered mandatory to use combustion conditions directly comparable between the two fuels and, at the same time, to maximize the possible comparisons (i.e., same air flow with different fuel flows, same fuel flow with different air flows, same global equivalence ratio and air flow with different fuels). In the extra-experiments we are planning we will try to reproduce the best way possible some of the already experimented burning conditions to have comparable results. Anyway, the set-up will not be exactly the same, since our experiments make use of the simulation chamber, and its exclusion can not be considered. Since our atmospheric chamber is currently engaged full time in non-postponable experiments, we will perform the request experiments as soon as possible, in agreement with the editor. We will be able to answer after the extra experiments requested by the referee.

Fourth, the authors should present SSA from the PAX instrument, to allow for a direct comparison of their measurements with the SSA reported by Moallemi et al. On line 386 the authors write "the comparison with previous literature (Moallemi et al., 2019) ... reported

the Single Scattering Albedo instead of the absorption coefficient". The authors cannot change how Moallemi et al. presented their data, but they can match their presentation to Moallemi et al's. The authors have SSA data and should present it.

Done.

Minor comments -----

The absorption coefficient divided by the number concentration is the absorption cross section. Please use this definition in Figure 11.

Modified and moved to Supplementary as requested by RC3.

What exactly is the difference between the MISG-1 and the MISG-2? Previous MISG characterization studies used the MISG-1.

We did not know the answer so we asked to the manufacturer. He explained us that there are just a few minor changes between MISG-1 and MISG-2. The main difference is that MISG-1 had a 1/2" exhaust tube while MISG-2 has a 3/4" exhaust tube. We thank Jason Olfert for this information.

Please specify the line length used between MISG and chamber. Obviously, this is important (see first major comment above). Please also specify the exhaust line length and i.d. (ideally pressure in the line would be reported, if that is not available then reporting these parameters will help).

We added the line length used between MISG and chamber in sect. 2.2. Since the exhaust line does not pass through the chamber volume, we believe that the only useful length is of the line between MISG and chamber.

We added in Line 159: The connection between MISG and ChAMBRe was made by Swagelok adaptors (size  $\frac{34}{}$ ) and ISO-K flanges (16 mm diameter) to avoid any possible leak; the length of the line was 65 cm.

Table 4. The dark red and dark purple look the same in a black-and-white printout. Use e.g. a lighter red.

**Done.**

The authors first mention Bischof et al. (2019) at line 292. The paper should be mentioned in the introduction; it is a characterization of the same MISG.

**Done.**

Line 79, the air flow is not internally split between combustion and carriage. This makes it sound like there are 2 divided flows. In fact there is one air flow, and some of the air is consumed for combustion.

**Corrected.**

Line 93, consider writing m3 air / m3 fuel (the unit m3/m3 is confusing.)

**To avoid confusion, we removed units as suggested by RC5.**

Line 104, Moore et al. demonstrated the relationship of stoichiometry with particle size for the miniCAST only. The miniCAST is unique from the MISG, because it is a quenched flame. The quenching height is fixed. In the MISG, the open tip can move up and down with fuel flow. Use a difference reference, or change the statement.

**We deleted the statement.**

Line 248. This is repeatability, not reproducibility.

**Corrected.**

Line 248, is the repeatability measured day-to-day? Between scans?

We added details in the text. The repeatability is measured between identical repeated experiments.

Figure 3. Missing error bars. Same for similar figures.

**Done.**

Figure 4 and line 292. The discussion of size vs. stoichiometry compares this work with prior work which was not performed at the same fuel flow rates. The comparison is not fair. It is more reasonable to conclude that the relationship changes at high fuel flow rates (if all data were plotted together, a trend might be observed). The authors need to reproduce earlier measurements to confirm their discussion, or change the discussion.

The Referee is right, the direct comparison is not possible since the feeding flows are different. Anyway, keeping in mind this information, we can discuss the trend of different particle properties by varying flows and compare them to the previous works. We added this statement (Line 292): "Even if the direct comparison between our findings and results from previous works are not directly comparable (since feeding flows are different), some similarities can be identified".

Line 307-316. The discussion compares "2 um" particles with "4 um" particles but the units are not the same. The 2 um was measured by TEM maximum dimension (or projected area?) and the 4 um was measured optically. Was the optical size corrected for the refractive index and shape of the particles? Please specify "projected diameter in an electron microscope" and "optical equivalent diameter". And please describe the calibration of the TSI OPS 3330 in Methods.

We used the default TSI refractive index. The OPS had been calibrated by the manufacturer. We appreciate the suggestion to uniform the quantities, but in our discussion we are not interested in giving precise numbers but just give the order of magnitude of the particle dimensions. Of course this parameter has been measured with very different techniques (TEM vs. OPS) and we are not even trying to match them. So we added in the text the specification of kind of diameter as suggested.

Figure 6. At what MISG flow rates were these data taken?

We added this information in the caption: "MISG was fuelled with 7 lpm of air and 75 mlpm of fuel during propane experiment or 127 mlpm of fuel during ethylene experiment. No cyclone."

Figure 7 and 8. Why not use units on the y axis if units are reported in the text? Please change to units. If the authors argue against units, then specify the maximum in the caption.

**Done.**

Figure 9 and 10. Please combine into Panel A and B of the same figure, to avoid repeating a long caption twice. The important point is that one used a cyclone.

**Done.**

Figure 9 and 10. Please change from "Relative EC concentration" to "EC:TC ratio" to make it clear what the EC is 'relative' to. The discussion mentions OC:EC as well, which is confusing. Please always use EC:TC and OC:TC.

Since it is not the EC:TC ratio but the EC concentration normalized to the highest concentration of the whole data set, we changed "Relative EC concentration" to "Normalized

EC concentration". We modified the discussion by using OC:TC.

Line 366-373. Blank and backing filters should be mentioned in Methods so that the reader is not surprised at the discussion here.

We added this mention in Sect. 2.5, line 214: "We also performed some tests adding a backup filter during the sampling to determine the volatile fraction of OC."

The authors should also mention that about 1000 ug/m3 EC was collected on the filters, which means that gas-phase VOCs become less important.

Added.

Anyway, is the discussion of OC correction relevant if only EC concentrations are reported? Isn't "PC" more relevant, since that is where biases can come in? (In other words, how difficult was it to determine the split point?)

We reported only EC concentration values because we observed that OC concentration values were negligible. Since sample contained almost only EC, thermograms were easy to analyze with split points clearly identifiable.

Figure 11. The authors normalized b\_abs to N\_SMPS. But in Figure 6 the authors showed that N\_OPS was important. Why did the authors ignore the particles that the OPS couldn't see? Was a cyclone used? Clarify the text and figure please.

The figure refers to experiments performed without the cyclone, we added the information in the text. We consider only SMPS data because the number concentration of super micrometric particles is negligible compared to the total number concentration. Anyway, when the cyclone was inserted between ChAMBRe and PAXs, particles generated from propane combustion were even more absorbent than the ethylene generated, even if with a small gap. As suggested by other referees, we moved this figure in the supplementary and we added the Figure with cyclone too.

Table 6. Was the AAE calculated using a power-law fit?

Yes

A fit to 3 points would not be reliable. I recommend reporting 2-wavelength calculations of the AAE, for blue-green and green-IR, (and optionally also blue-IR) which also allows the reader to observe the consistency between the individual PAX instruments.

We added a Table in the Supplementary with the results of AAE from the 2-wavelenght calculations.

Figure 12-14. Consider using open/closed symbols to enhance readability in black-and white printouts.

Done.

---

## Author Comment (AC5)

Atmos. Meas. Tech. Discuss., referee comment RC4
https://doi.org/10.5194/amt-2021-345-RC4, 2021

[Figure]

**Addendum to RC2 review**

Anonymous Referee #2
* * *
Referee comment on "Characterization of the MISG soot generator with an atmospheric simulation chamber" by Virginia Vernocchi et al., Atmos. Meas. Tech. Discuss., https://doi.org/10.5194/amt-2021-345-RC4, 2021
* * *
Addendum to review of 10.5194/amt-2021-345, Characterization of the MISG soot generator with an atmospheric simulation chamber, by Vernocchi et al.

My original review recommended publication of the manuscript by Vernocchi et al. primarily for its new data on supermicron aggregates in terms of optical particle size and optical properties. This recommendation stands.

This addendum adds to my earlier review to clarify three minor points regarding the manuscript by Kazemimanesh et al. (K2018), which is the only previous manuscript to study superaggregates from MISG ethylene flames.

The points are as follows:

1)      My original review stated that K2018 reported TEM size distibutions up to 2 um. This is true, but K2018 also reported aerodynamic size distibutions. (Moallemi et al. 2018 reported only TEM.) The physical interpretation of aerodynamic and optical size distributions should be discussed in detail (see e.g. https://dx.doi.org/10.1080/027868290903907). What is the optical equivalent diameter of 2 um aerodynamic diameter soot aggregates in the supermicron regime, considering morphology? Calculating the answer to this question is difficult, but measuring it is simple: the authors can compare OPS size distributions with/without the cyclone. (This comment extends one of my original minor comments.)

Since our atmospheric chamber is currently engaged full time in non-postponable experiments, we will perform the request experiments as soon as possible, in agreement with the editor. We'll try our best to characterize these super-aggregates, using the instruments we have in our lab and within the scope of the present work.

2)      K2018 discussed superaggregate formation in a stagnation plane, citing literature by Chakrabarty et al. different to the citation I gave earlier. The stagnation plane hypothesis is inconsistent with the present manuscript's hypothesis that coagulation occurred in the sampling lines. The stagnation plane hypothesis may also better explain the difference in EC:TC of the superaggregates. Regardless, I still recommend that the authors test different sampling line lengths directly since that test is simple. (This comment extends my original 2nd major comment.)

We have planned these experiments in the next weeks. We'll try our best considering our comments in the original review of the Referee 2.

3)      K2018 also showed that superaggregate formation depends on fuel flow rate, with negligible superaggregates observed at the lowest flow rate (which also produced a lower number concentration). So did the authors observe 'larger' superaggregates because they used a higher fuel flow rate, or because they used an optical particle sizer instead of an aerodynamic one? (This comment extends my original 1st and third comments.)

We thank the Referee for the thorough speculation on the super-aggregates origin. We don't have the answer to this question, but we can try a simple experiment: we will use lower flow rates to see how their dimension change with them.

---

## Author Comment (AC6)

Atmos. Meas. Tech. Discuss., referee comment RC5
https://doi.org/10.5194/amt-2021-345-RC5, 2021 ©
Author(s) 2021. This work is distributed under the Creative
Commons Attribution 4.0 License.

[Figure]

**Comment on amt-2021-345**

Anonymous Referee #4

Referee comment on "Characterization of the MISG soot generator with an atmospheric simulation chamber" by Virginia Vernocchi et al., Atmos. Meas. Tech. Discuss., https://doi.org/10.5194/amt-2021-345-RC5, 2021

**Review of "Characterization of the MISG soot generator with an atmospheric simulation chamber"**

We performed some additional experiments useful to reply at specific Referee questions.

**General comments:**

Page 10 – Fig. 4 and the discussion around it: The particle mode diameter reported for ethylene flames is constant at ~175 nm. This is inconsistent with previously reported values of ~240 nm and up to 270 nm (Kazemimanesh et al., 2019). The same reference also reported an initial sharp increase in particle size and concentration with increasing ethylene flow rate, which eventually levelled off to a relatively constant value. This is in contrast to the trend seen in this paper. These differences must be noted and discussed in the paper.

We added the discussion about these differences, that probably depended on the different combustion conditions.

Line 292: Even if the direct comparison between our findings and results from previous works (Bischof et al., 2019; Kazemimanesh et al., 2019; and Moallemi et al., 2019) are not directly comparable (since feeding flows and global equivalence ratios are different), some similarities can be identified. Previous works observed that by increasing the fuel flow, the particle number concentration increases too, that is what we observed for propane. In addition, Bischof (2019) also reported that the particle mode diameter, with propane, did not depend on the global equivalence ratio, as we also observed, but for ethylene. Kazemimanesh (2019) showed a clear increase in mode diameter, corresponding to an increase of fuel flow rate, that reached a quite constant value (i.e., around 240-270 nm) for ethylene. This trend differs from our observations, since the mode diameter in our case turned out to be quite stable at about 175 nm independently on feeding flows. This difference is probably due to the global equivalence ratios used: while in (Kazemimanesh et al., 2019) global equivalence ratios are lower than 0.206, in our case they are higher than 0.213. In (Moallemi et al., 2019), instead, they observed an opposite behaviour for mode diameters: they retrieved that at fixed fuel flow, a higher air flow produced a slight decrease of the mode diameter. Both (Moallemi et al., 2019) and (Bischof et al., 2019) measured mode diameters < 200 nm, but they used different combustion conditions (i.e., lower global equivalence ratios resulting from higher air flow or lower fuel flow). We can conclude that, as

expected, global equivalence ratio is the principal parameter affecting size distributions of soot particles.

Anyway, as request by RC2, we carried out experiments that replicate some of the conditions used in the previous works, so we will able to compare the same operative conditions used by (Kazemimanesh et al., 2019).

We replicated the following conditions: 9 lpm of air - 100 mlpm of fuel (ethylene) and 10 lpm of air - 100 mlpm (ethylene).

*Table 1: Comparison between results of previous literature work and our replicated experiments.*

|  | Kazemimanesh et al., 2019 | This work |
|---|---|---|
|  | Mode diameter (nm) | Mode diameter (nm) |
| Ethylene: 9 - 100 | 242 | 191 ± 8 |
| Ethylene: 10-100 | 250 | 220 ± 9 |

In the revised text, we added:

Line 267: In addition, we reproduced some of the conditions investigated in the previous works obtaining a good agreement for the mode diameter and SSA figures (see Supplementary for details §3).

Line 498: The formation of superaggregates is related to high particle concentration in the exhaust line. This means that by diluting the MISG exhaust, the formation of these large aggregates can be alleviated. Kazemimanesh et al. (2019) and Chakrabarty et al. (2012) suggest that these superaggregates are formed at the stagnation plane of the flame tip, which seems more plausible. The authors should note and discuss these differences in the paper (not in the conclusions section).

This point has been raised by more than one Referee, and we agree that it is an interesting point to investigate. We will be able to answer to this question and add the results in the revised text after some extra experiments, by inserting a diluter between MISG and ChAMBRe as suggested.

We diluted the MISG exhaust just before the outlet of the generator, by adding an extra air flow, the ratio between dilution air and MISG generator was 4:1. No significant differences were observed in the super-micrometric range, suggesting that superaggregates are formed at the stagnation plane of the flame tip, as reported in Kazemimanesh et al. (2019) and Chakrabarty et al. (2012).

[Figure]

*Figure 1: Comparison between mass size distributions measured by SMPS and OPS. The MISG was fuelled with 7 lpm of air and 127 mlpm of ethylene.*

In the revised text, we added:

Line 309: ethylene combustion produced a limited number of big particles, likely super-aggregates, probably formed at the stagnation plane (Chakrabarty et al., 2012). This hypothesis was confirmed by dedicated experiments with the setup specifically modified in respect to the basic one (see Supplementary Fig. S.2).

---

## Author Comment (AC7)

Atmos. Meas. Tech. Discuss., referee comment RC2
https://doi.org/10.5194/amt-2021-345-RC2, 2021 ©
Author(s) 2021. This work is distributed under the Creative
Commons Attribution 4.0 License.

[Figure]

**Review of Vernocchi et al. MISG characterization**

Anonymous Referee #2
* * *
Referee comment on "Characterization of the MISG soot generator with an atmospheric simulation chamber" by Virginia Vernocchi et al., Atmos. Meas. Tech. Discuss., https://doi.org/10.5194/amt-2021-345-RC2, 2021 and https://doi.org/10.5194/amt-2021-345-RC4, 2021
* * *
Review of 10.5194/amt-2021-345, Characterization of the MISG soot generator with an atmospheric simulation chamber, by Vernocchi et al.

We performed the additional experiments requested by the Referee.

That study also used a different flow rate. Given this emphasis I would like to request one additional experiment is made before publication. The authors should directly test their hypothesis that "super-aggregates...are likely formed directly in the exhaust line where particles density is very high" (lines 498-499). If this is the case, then could the issue be solved simply by diluting immediately after the MISG? The experiment would be simple. The authors need only to run the MISG with 3 line lengths. Very short, normal (as used previously), and very long. For each line length, measure with the OPS and SMPS. The results should be reported as combined OPS-SMPS size distributions in mass and number weighting.

2) K2018 discussed superaggregate formation in a stagnation plane, citing literature by Chakrabarty et al. different to the citation I gave earlier. The stagnation plane hypothesis is inconsistent with the present manuscript's hypothesis that coagulation occurred in the sampling lines. The stagnation plane hypothesis may also better explain the difference in EC:TC of the superaggregates. Regardless, I still recommend that the authors test different sampling line lengths directly since that test is simple. (This comment extends my original 2nd major comment.)

This point has been raised by more than one Referee, and we agree that it is an interesting point to investigate. We will be able to answer to this question after some extra experiments, 1) by modifying the line length as suggested, 2) by inserting a dilution system just after the SG exhaust. Anyway, we have just a doubt about the effect produced by the modification of line length. After the small quartz cell where the flame burns, the exhaust is carried outside the SG after passing through a copper serpentine, with length roughly 40 cm long. If coagulation happens in this section, no way to understand if super-aggregates are formed in the flame or after.

We tested different line lengths: the short line was 30 cm, the normal line was 65 cm long and the long line was about 5 m. We also diluted MISG exhaust just after the outlet of the generator

maintaining the normal length of the exhaust line, by adding an extra air flow, the ratio between dilution air (dry) and MISG generator was 4:1.

Only the experiment with the longest line showed a significant decrease in particle concentration, probably due to the losses inside the pipeline. These results suggest that superaggregates were formed at the stagnation plane of the flame tip, as correctly reported by the Referee and references he cited (Kazemimanesh et al. (2019) and Chakrabarty et al. (2012)).

[Figure]

*Figure 1: Comparison between mass size distributions measured by SMPS and OPS. The MISG was fuelled with 7 lpm of air and 127 mlpm of ethylene.*

[Figure]

*Figure 2: Comparison between number size distributions measured by SMPS and OPS. The MISG was fuelled with 7 lpm of air and 127 mlpm of ethylene.*

In the revised text, we added:

Line 309: ethylene combustion produced a limited number of big particles, likely super-aggregates, probably formed at the stagnation plane (Chakrabarty et al., 2012). This hypothesis was confirmed by dedicated experiments with the setup specifically modified in respect to the basic one (see Supplementary Fig. S.2).

Third, I would also request that the authors try harder to reproduce exactly the conditions used in previous studies. As it is, the authors have used higher fuel flow rates (equivalence ratios) than all previous studies (Kazemimanesh et al., 2019; Moallemi et al, 2019; Bischof et al., 2019). It is unclear to me why the authors have not reproduced previous measurements exactly, to allow for comparable results. Is it because the authors used long line lengths and changed the pressure downstream of the flame? Is it because the authors used an "MISG-2" and not an "MISG-1"? Also, as mentioned above, the abstract should emphasize this difference in flow rate.

The opportunity to reproduce exactly the condition used in previous works is for sure a good scientific procedure. However, we did not reproduce previous measurements for several reasons. First, our intention was to explore new operation conditions in order to expand the knowledge of the SG. Secondary, we wanted to compare the soot produced by propane and ethylene, since all the previous works focused on one fuel only at a time. So, we considered mandatory to use combustion conditions directly comparable between the two fuels and, at the same time, to maximize the possible comparisons (i.e., same air flow with different fuel flows, same fuel flow with different air flows, same global equivalence ratio and air flow with different fuels). In the extra-experiments we are planning we will try to reproduce the best way possible some of the already experimented burning conditions to have comparable results. Anyway, the set-up will not be exactly the same, since our experiments make use of the simulation chamber, and its exclusion can not be considered.

We replicated some of the combustion conditions reported in the previous literature works. We explored 9 lpm of air - 100 mlpm of fuel and 10 lpm of air - 100 mlpm of fuel for ethylene and 8 lpm of air - 61 mlpm of fuel and 9 lpm of air - 61 mlpm of fuel for propane.

*Table 1: Comparison between results of previous literature work and our replicated experiments.*

| | Kazemimanesh et al., 2019 | | This work | |
|---|---|---|---|---|
| | Mode diameter (nm) | | Mode diameter (nm) | |
| Ethylene: 9 - 100 | 242 | | 191 ± 8 | |
| Ethylene: 10-100 | 250 | | 220 ± 9 | |
| | Moallemi et al., 2019 | | This work | |
| | Mode diameter (nm) | SSA | Mode diameter (nm) | SSA |
| Propane: 8 - 61 | 150 - 190 | 0.17 – 0.22 | 202 ± 12 | 0.16 |
| Propane: 9 -61 | 130 - 160 | 0.16 – 0.20 | 165 ± 10 | 0.14 |

In the revised text, we added:

Line 267: In addition, we reproduced some of the conditions investigated in the previous works obtaining a good agreement for the mode diameter and SSA figures (see Supplementary for details §3).

1) My original review stated that K2018 reported TEM size distibutions up to 2 um. This is true, but K2018 also reported aerodynamic size distibutions. (Moallemi et al. 2018 reported only TEM.) The physical interpretation of aerodynamic and optical size distributions should be discussed in detail (see e.g. https://dx.doi.org/10.1080/027868290903907). What is the optical equivalent diameter of 2 um aerodynamic diameter soot aggregates in the supermicron regime, considering morphology? Calculating the answer to this question is difficult, but measuring it is simple: the authors can compare OPS size distributions with/without the cyclone. (This comment extends one of my original minor comments.)

Since our atmospheric chamber is currently engaged full time in non-postponable experiments, we will perform the request experiments as soon as possible, in agreement with the editor. We'll

try our best to characterize these super-aggregates, using the instruments we have in our lab and within the scope of the present work.

We measured SMPS+OPS distributions both without and with the cyclone as suggested, for 7 lpm of air and 127 mlpm of ethylene.

[Figure]

*Figure 3: Comparison between number size distributions measured by SMPS and OPS. The MISG was fuelled with 7 lpm of air and 127 mlpm of ethylene.*

3) K2018 also showed that superaggregate formation depends on fuel flow rate, with negligible superaggregates observed at the lowest flow rate (which also produced a lower number concentration). So did the authors observe 'larger' superaggregates because they used a higher fuel flow rate, or because they used an optical particle sizer instead of an aerodynamic one? (This comment extends my original 1st and third comments.)

We thank the Referee for the thorough speculation on the super-aggregates origin. We don't have the answer to this question, but we can try a simple experiment: we will use lower flow rates to see how their dimension change with them.

We performed an experiment using 6 lpm of air and 80 mlpm of ethylene. The formation of super-aggregates larger than 4 µm decreased consistently. Answering to the Referee's question, we observed larger superaggregates because we used a higher fuel flow rate.

[Figure]

*Figure 4: Comparison between mass size distributions measured by SMPS and OPS.*

In the revised text, we added:

Line 317: Anyway, super-aggregates formation by ethylene combustion can be partly reduced by using lower air and fuel flow rates (see Supplementary Fig. S.3 for example).